# AdriE: a high-resolution ocean model ensemble for the Adriatic Sea under severe climate change conditions

Davide Bonaldo[1], Sandro Carniel[2], Renato R. Colucci[2,a], Cléa Denamiel[3,4], Petra Pranić[5], Fabio Raicich[6], Antonio Ricchi[7,8,1], Lorenzo Sangelantoni[9], Ivica Vilibić[3,4], and Maria Letizia Vitelletti[1]

[1]National Research Council of Italy, Institute of Marine Sciences (CNR-ISMAR), Venice, Italy
[2]National Research Council of Italy, Institute of Polar Sciences, (CNR-ISP), Venice, Italy
[a]previously at: CNR-ISMAR, Trieste, Italy
[3]Ruder Bošković Institute, Division for Marine and Environmental Research, Zagreb, Croatia
[4]Institute of Adriatic Crops and Karst Reclamation, Split, Croatia
[5]Institute of Oceanography and Fisheries, Split, Croatia
[6]National Research Council of Italy, Institute of Marine Sciences (CNR-ISMAR), Trieste, Italy
[7]Department of Physical and Chemical Sciences, University of L'Aquila, L'Aquila, Italy
[8]Center of Excellence in Telesensing of Environment and Model Prediction of Severe Events (CETEMPS), University of L'Aquila, L'Aquila, Italy
[9]CMCC Foundation - Euro-Mediterranean Center on Climate Change, Bologna, Italy

**Correspondence:** Davide Bonaldo (davide.bonaldo@cnr.it)

**Abstract.** The Adriatic Sea (Eastern Mediterranean basin) is traditionally considered as a natural laboratory for studying a number of oceanographic processes of global interest, including coastal dynamics, dense water formation and thermohaline circulation. More recently, the intensification of the effects of climate change and the increasing awareness of its possible consequences on the natural and socio-economic assets of the Adriatic basin have opened new research questions and reframed most of the existing ones into a multi-decadal time scale. In this perspective, a description of the possible evolution of the physical oceanographic processes is one of the key requirements for addressing the multi-disciplinary challenges set by climate change, but up to now it has not been possible to combine, for this basin, a sufficiently high resolution in the process description with an estimate of the uncertainty associated with the predictions.

This work presents an ensemble modelling approach (AdriE - Adriatic Sea Ensemble) for the kilometre-scale description of hydrodynamics in the Adriatic Sea in an end-of-century time frame. Addressing 3D circulation and thermohaline dynamics within the Regional Ocean Modelling System (ROMS), the ensemble consists of six climate runs encompassing the period from 1987 to 2100 in a severe RCP8.5 scenario forced by the SMHI-RCA4 Regional Climate Model, driven by as many different CMIP5 General Climate Models made available within the EURO-CORDEX Initiative. The climate ensemble is flanked by a dedicated evaluation run for the 1987-2010 period, in which SMHI-RCA4 has been driven by reanalysis fields approximating the best available boundary conditions, thus isolating the intrinsic sources of uncertainty of the RCA4-ROMS modelling chain. In order to allow a direct comparison, the assessment of the model skills in the evaluation run borrows, as far as possible, data and approaches used for the evaluation of a recent kilometre-scale, multi-decadal modelling effort for this region. The model performances are mostly aligned with the state-of-art reference. In particular, good results in describing the main features of Marine Heat Waves and Cold Spells, such as timing, intensity, and interannual variability, indicate that the AdriE ensemble

can effectively be used for studies on the occurrence and effects of thermal extremes in the basin. Future projections suggest an increase in temperature and salinity at upper and intermediate depths, resulting in an overall decrease in water density and possibly in deep ventilation rates. Projected variations are stronger in summer and autumn, and in these seasons the ensemble range is larger than the spatial variability of the quantities and occasionally comparable with the intensity of the climate signal, highlighting the importance of an ensemble approach to treat the climate variability at this time scale. The dataset presented in this study, which can be used for the analysis of coastal and continental margin processes of general interest, is fully available upon request to the corresponding author, and monthly averages of the main quantities are available for each run on a dedicated Zenodo Repository.

## 1  Introduction

The Adriatic Sea is a semi-enclosed sub-basin of the eastern Mediterranean Sea, elongating along the NW-SE direction and surrounded along its major axis (approximately 800 km) by the Apennines and the Balkans, and closed on its northern end by the Padan-Venetian-Friulian plain. The basin is characterised by a broad and shallow continental shelf, crossed along the SW-NE direction at approximately 43 °N by the Jabuka Pit (maximum depth 280 m) and eventually plunging into the southern Adriatic Pit (SAP) between the Apulia peninsula and the Montenegro-Albanian coast. The SAP, whose depth ranges between 180 m at the continental shelf edge and 1200 at its deepest point, is connected to the Ionian sea through the Otranto strait and its 780-metre deep sill (Orlić et al., 1992; Bonaldo et al., 2016).

Due to the coexistence of manifold meteo-oceanographic processes (e.g. river plumes and coastal fronts, storm surges, seiches, and meteotsunamis, dense water formation and cascading) and different socio-economical pressures and interests, the Adriatic Sea is traditionally regarded as a natural oceanographic laboratory and a paradigm for diverse applications. For instance, the presence of highly-exposed sites of outstanding natural and cultural value and long stretches of low-lying sandy beaches has been a key motivation for studies on hazard factors such as coastal erosion (Bonaldo et al. 2019, and references therein) and flooding (Vilibić et al., 2017), and on the physical drivers of these events. Furthermore, the challenges related to the basin morphology and its orographic configuration have motivated numerical modelling and Earth Observation analysis developments for coastal regions (Sanchez-Arcilla et al., 2021; Umgiesser et al., 2022). The importance of the Adriatic Sea as a cold engine for the Mediterranean thermohaline circulation (Bergamasco and Malanotte-Rizzoli, 2010) has fostered fecund research lines on dense water formation, deep ventilation, and the relations of these processes with continental margin geomorphology and deep-sea ecology (Bonaldo et al., 2016; Bargain et al., 2018; Vilibić et al., 2023). In recent years, climate change and its effects have progressively gained prominency in this scene, increasingly calling for the projection of the possible evolution of marine and coastal systems in this basin over a multi-decadal time scale at a sufficient level of detail. While the predominantly coastal setting of the Adriatic basin demands a high-resolution modelling approach, the temporal range of the processes related to climate change requires a multi-decadal coverage. In this direction some efforts have been undertaken at first for wind waves and barotropic dynamics (Benetazzo et al., 2012; Lionello et al., 2012; Bonaldo et al., 2020). Subsequently, recent achievements in terms of high-resolution, three-dimensional ocean modelling were reached by the AdriSC modelling suite

(Denamiel et al., 2019; Pranić et al., 2021; Denamiel et al., 2021a), in which kilometre-scale projections of trends, variability and extremes in both atmosphere and ocean have been achieved (Tojčić et al., 2024), but only for one climate scenario (RCP 8.5) and for far-future period (2070-2100). At the scales considered in those works, running a climate model requires enormous computational resources, thus the full scenario simulation was not affordable and pseudo-global warming approach has been implemented and extended to the ocean (Denamiel et al., 2020). Nonetheless, climate change projections are subject to several levels of uncertainty, from the evolution of the global climate to how the signal propagates through different scales, and how the adopted numerical description affects the final results. This uncertainty is typically addressed by means of ensemble modelling approaches in which the some degrees of freedom are explored by dedicated model runs, but the computational demand of such a task is typically unaffordable outside of a dedicated consortium. On the other hand, such coordinated efforts typically address relatively large spatial scales, failing to capture the local features of climate change. The Med-CORDEX initiative (www.medcordex.eu), although promising over the long shot, is still not providing extensive high-resolution, end-of-century model fields for the Adriatic region, and some shortcomings have been pointed out in the representation of the main thermohaline processes, and particularly in dense water formation (Dunić et al., 2019, 2022). The goal of the present work is to bridge this gap for the Adriatic basin, proposing a trade-off between the very-high resolution required to properly reproduce the local oceanographic processes and the need for an estimate of the uncertainty associated with the future predictions. In this paper we introduce a six-member ocean model ensemble for the Adriatic Sea up to the end of century in the severe climate change scenario RCP8.5 (Church et al., 2013), assessing its performance and discussing the potential and the possible limitations for its applicability in the analysis of the physical oceanographic processes of the basin as well as in downscaling applications for local studies or as a source of information for multidisciplinary efforts.

## 2   Materials and methods

### 2.1   AdriE ensemble model implementation

The AdriE ensemble is based on a set of implementations of the ROMS modelling system (Haidvogel et al., 2008) over the Adriatic Sea, with an open boundary at the Otranto Strait (Figure 1a). All simulations are forced with 0.11-degree resolution, 6-hourly atmospheric fields (total cloud fraction, relative humidity, sea-level air pressure, precipitation, long- and shortwave radiation, near-surface air temperature, and near-surface wind velocity components) from the SMHI-RCA4 regional climate model (RCM) (Samuelsson et al., 2011) driven by six Global Climate Models (GCMs) from the CMIP5 initiative (Taylor et al., 2012) and listed in Table 1. SMHI-RCA4 has been used as a reference model in consideration of its extensive record of regional-scale climate projections and its performances in reproducing relevant climate processes (which have been found well representative of the CORDEX multi-model ensemble variability, see Kotlarski et al. 2014; Coppola et al. 2021; Diez-Sierra et al. 2022; Vautard et al. 2021), as well as of the availability of the relevant variables with the necessary sub-daily temporal resolution for a comparatively large number of simulations corresponding to different GCMs. The climate ensemble consists of six free-running climate simulations bracketing the period 1987-2099, thus including part of the "historical" (1987-2005, based on the observed radiative forcing) period, and future "scenario" (2006-2099, based on projected values of radiative

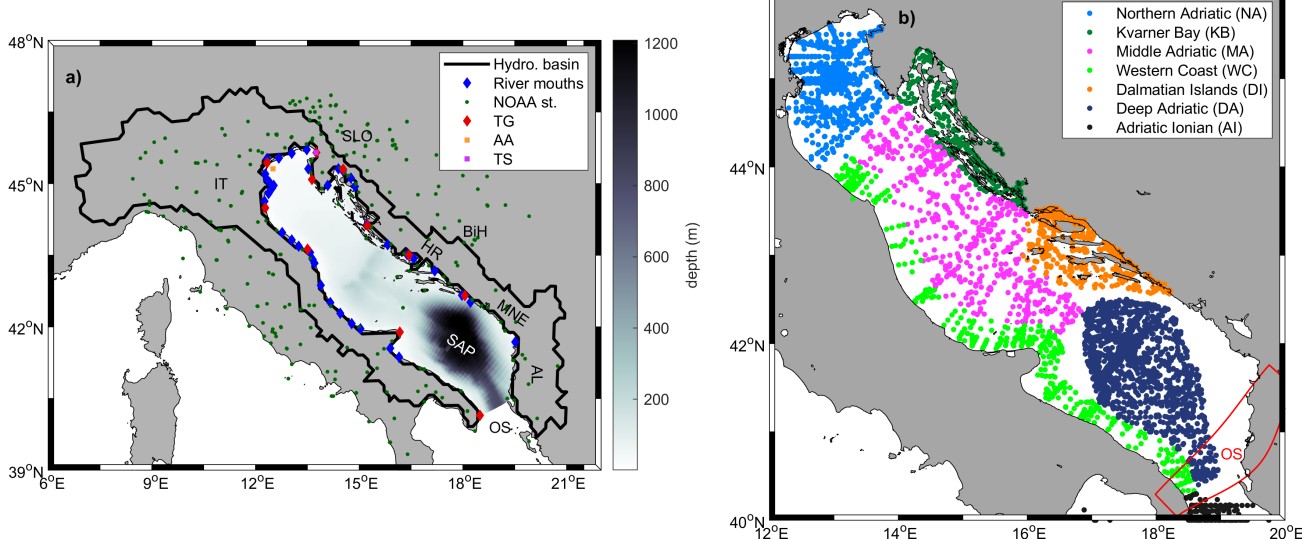

**Figure 1.** Geographical setting. a) Model domain and bathymetry of the Adriatic Sea, with indication of the NOAA stations and of the tide gauges (TG) considered in the validation, as well as the position of the Acqua Alta (AA) and Trieste (TS) monitoring stations. The thick black line indicates the hydrographic basin draining into the AS, and the blue diamonds represent the position of the freshwater inputs included in the model. OS represents the open boundary at the Otranto Strait. b) Position of the CTD measurements and identification of the subdomains described by Pranić et al. (2021), namely Northern Adriatic (NA); Kvarner Bay (KB); Middle Adriatic (MA); Western Coast (WC); Dalmatian Islands (DI); Deep Adriatic (DA); Adriatic Ionian (AI). The region enclosed within the red line represents an additional subdomain (Otranto Strait - OS) for the evaluation in the surroundings of the model domain boundary.

forcing) under the severe RCP8.5 pathway. In this work, the comparison between end-of-century and recent past conditions is carried out by comparing climate runs in the scenario (SCE, 2070-2099) and control (CTR, 1987-2016) periods. This set of simulations is flanked by an "evaluation" run (EV hereinafter) in which SMHI-RCA4 was driven by the ERA-INTERIM reanalysis (Dee et al., 2011) throughout the 1987-2010 period and therefore synchronised, unlike the climate simulations, with the observed day-to-day climate variability. This allows to isolate the RCM-ROMS modelling chain and assess its skills under an approximation of the best available information at the boundary (ideally the "perfect boundary conditions", see Christensen et al. 1997), and permits a one-to-one comparison between modelled and observed records (that is, modelled fields for each day at each location should match as closely as possible the observations collected in the same days and locations). Conversely, since GCMs are free running models not constrained, once initialised, to the observed state of the climate system components, the analysis of the climate runs allows the assessment of the skills of the whole modelling chain only in terms of aggregated statistical properties: as a consequence, the methodology for assessing reanalysis-driven evaluation runs does not fully apply to GCM-driven climate simulations.

Table 1 summarises the main features of the runs performed. To characterise the largest spectrum of uncertainty we opted for maximising the number of GCMs since these exert a larger influence than RCMs on climate change signal variability

**Table 1.** Overview of climate and evaluation runs and driving models for RCM SMHI-RCA4 from EURO-CORDEX. EV and EV* are forced by the same model but distinguished by the use of different versions of the product from which the boundary conditions were retrieved.

| Run | Driving GCM | Period | leap years |
|---|---|---|---|
| NCC | NCC-NorESM1-M r1i1p1 | 1987-2099 | N |
| CNRM | CNRM-CERFACS-CNRM-CM5 r1i1p1 | 1987-2099 | Y |
| IPSL | IPSL-IPSL-CM5A-MR r1i1p1 | 1987-2099 | N |
| MPI | MPI-M-MPI-ESM-LR r1i1p1 | 1987-2099 | Y |
| ICHEC | ICHEC-EC-EARTH r12i1p1 | 1987-2099 | Y |
| ICHECb | ICHEC-EC-EARTH r3i1p1 | 1987-2099 | Y |
| EV | ECMWF-ERAINT | 1987-2010 | Y |
| EV* | ECMWF-ERAINT | 1987-2010 | Y |

(Christensen and Kjellström, 2020, 2022). The use of different models entails a broad spectrum of uncertainty sources (different formulations, numerical schemes, dynamical features, etc.), while different realizations of the same model ("r" index) address different, but equally realistic, initial conditions.

The horizontal discretisation follows the orthogonal curvilinear grid used for the wave climate projections described by Bonaldo et al. (2020), composed of $75 \times 180$ nodes with resolution ranging from approximately 2 km in the northern regions
and progressively coarsening to nearly 10 km in the southeasternmost end of the domain, whereas the water column is discretised into 15 terrain-following $\sigma$-levels with increasingly larger thickness towards the bottom. Six sub-domains were identified consistently with the work by Pranić et al. (2021), (namely Northern Adriatic, Kvarner Bay, Middle Adriatic, Western Coast, Dalmatian Islands, and Deep Adriatic; the Adriatic-Ionian subdomain lies mostly outside of the model domain and was not considered in this study), to facilitate a direct comparison, based on their thermohaline properties and circulation features (Figure 1b). An additional subdomain (Otranto Strait) was also introduced in the southeastern five-cell buffer in order to verify the
model performance close to the open boundary. The bathymetry was reinterpolated from finer-resolution information adopted in previous works (Benetazzo et al., 2014; Bonaldo et al., 2016, 2019) and smoothed as described by Sikirić et al. (2009) in order to prevent the appearance of artifact horizontal pressure gradients and consequent circulation features.

Potential temperature ($\theta$), salinity ($S$), momentum and sea level were prescribed as boundary conditions at the Otranto
Strait and derived from the MFS 1/16° (Simoncelli et al., 2019). Chapman conditions (Chapman, 1985) were imposed for free surface, Flather conditions (Flather, 1976) for 2D momentum components, and nudged radiative conditions for 3D momentum components, $\theta$, and $S$. For the EV run, daily reanalysis values were interpolated on the model grid points throughout the cross section. For the climate runs, climatological monthly values were first computed from the reanalysis fields with reference to the 1987-2017 period. Potential temperature and salinity values were then perturbed with the anomalies computed, with reference
to the same period, from Med-CORDEX derived CMCC-CM profiles (Scoccimarro et al., 2011) in the northeasternmost grid

cell of the Ionian Sea. Furthermore, since during the post-processing of the climate runs a new version of MFS was released (1/24° horizontal resolution, see Escudier et al. 2020), an additional evaluation run EV* was carried out to assess the impact of the difference between the two versions on the description of the Adriatic Sea dynamics and the possible implications for the climate projections. Time-averaged potential temperature, salinity, and velocity patterns and their trends are summarised in Figure 2. The thermohaline properties distribution appears consistent with typical values of the Eastern Intermediate Water (EIW, Schroeder et al. 2024), particularly in terms of the component identified in previous literature as Levantine Intermediate Water (LIW, also indicated in the Southern Adriatic Sea as "Modified" LIW - MLIW), entering the basin along the eastern continental margin (see for instance Bonaldo et al. 2016, and references therein). The known cyclonic flow across the boundary cross section is weaker in EV than in EV*, but this does not affect the capability to reproduce the main circulation patterns in the southern Adriatic basin, as will be shown in Section 3. The multidecadal trends for potential temperature in the climate simulations in the CTR period is well bracketed between the values from the EV and EV* runs (namely, the CMEMS reanalyses used as boundary conditions) and consistent (at least for surface values) with the estimates by Juza and Tintoré (2021), while salinity trends appear weaker and less consistent both in the evaluation and in the climate runs.

River mouths were included as 39 point freshwater sources. Climatological values from Raicich (1994) were used for the main Alpine (Isonzo, Tagliamento, Piave, Brenta, Adige, and Po split into five branches) and Apennine (Reno, Foglia, Metauro, Esino, Musone, Potenza, Chienti, Tronto, Pescara, Sangro, Trigno, Biferno, Cevaro, Ofanto) rivers, as well as for the Drin in the south-eastern Balkans, while the freshwater input from the Venice lagoon drainage basin was taken from Zuliani et al. (2005). For the rivers and submarine springs along the Croatian coast (Mirna, Raša, Rječina, Bakarac, Crikvenica, Zrmanja, Krka, Jadro, Cetina, Neretva, Ombla, and the Senj and Dubrovnik/Kupari hydropower plants) estimates were taken from Janeković et al. (2014), which in turn also relied on Raicich (1994) for the western coast. For the EV and EV* runs, the input from the Po river was computed from the observed discharge at Pontelagoscuro (approximately 80 km upstream, data publicly available at https://simc.arpae.it/dext3r/, SI@SIMC@ARPAE 2022), whereas for the climate runs climatological river discharges were rescaled following the spatially-averaged modelled rainfall anomaly on the Adriatic catchment (Figure 1a). Tides were included only in the EV and EV* runs by imposing 15 tidal constituents from the TPXO dataset (Egbert and Erofeeva, 2002) on the open boundary at the Otranto Strait.

## 2.2 Observational datasets and validation approach

The evaluation of the model skills was based on a broad and heterogeneous set of observations for different variables and from different sources (Table 2). In order to maximise the comparability of the results, and as far as it was relevant and consistent with the aim of this study, the analysis used the datasets, quantities and methods from the works by Denamiel et al. (2021a) and Pranić et al. (2021).

E-OBS v21.0 (Cornes et al., 2018) provided gridded data with 0.1-degree resolution for daily rainfall (R), sea-level pressure (SLP), and near-surface temperature (T) over land. Delivered and periodically updated by the Copernicus Climate Change Service, this dataset combines in-situ observations ensembles and is used as a reference for the evaluation of RCMs within the EURO-CORDEX community (Kotlarski et al., 2014; Varga and Breuer, 2020). Additional rainfall data were also collected

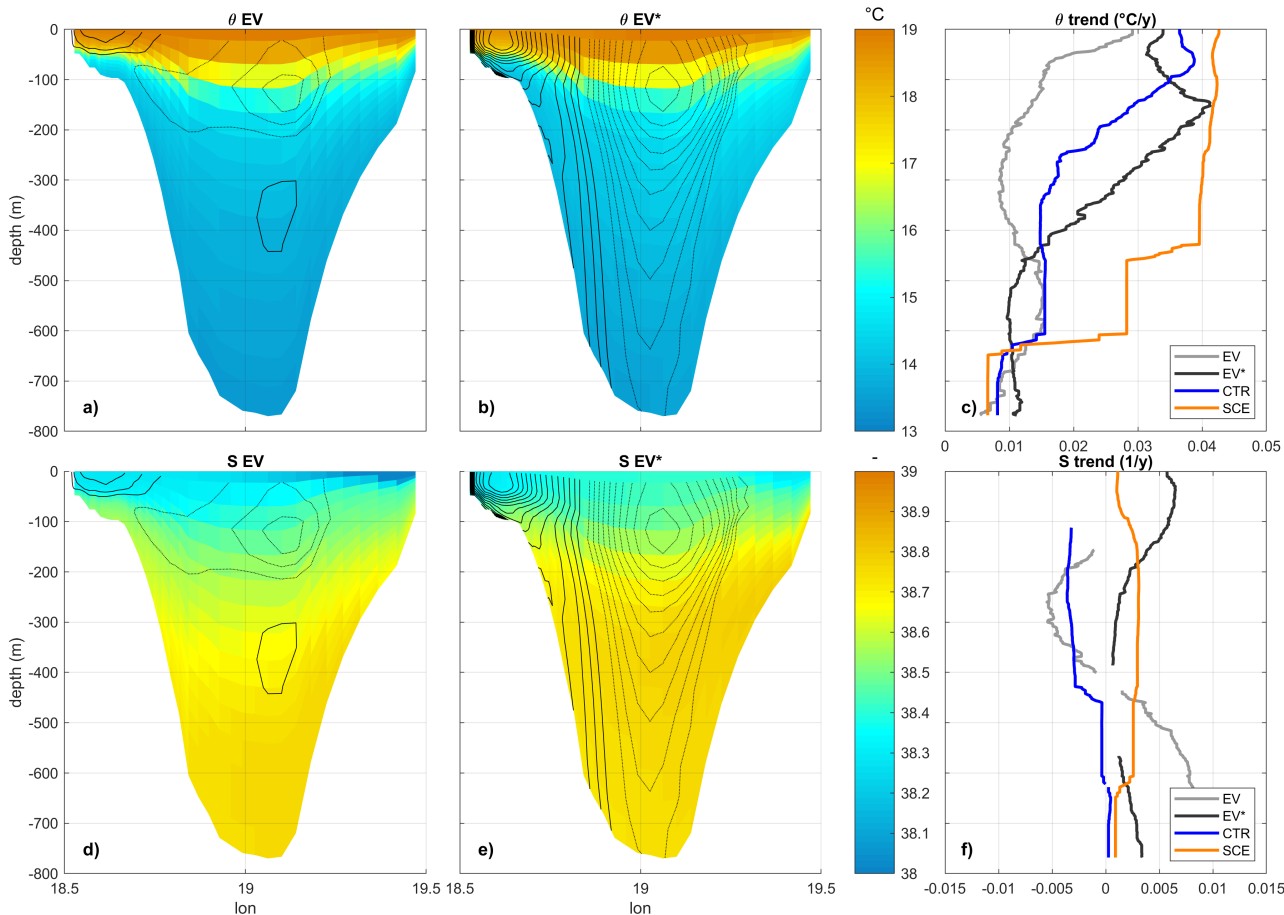

**Figure 2.** Time-averaged boundary conditions for potential temperature (panels a, b), salinity (d, e), and cross-transect velocity (0.01 m s$^{-1}$ contours in panels a, b, d, and e, thick and dashed lines representing respectively outflow and inflow) for EV and EV* runs, and trend profiles (blank values where not statistically significant) for evaluation and climate runs in different periods (c, f).

from the Tropical Rainfall Measuring Mission (TRMM) gridded dataset (Huffman et al., 2016) with 0.25-degree resolution over land and sea, based on the combination of microwave-IR data calibrated with rain gauges over land. Gridded near-surface wind information (zonal and meridional components -U, V- and intensity UV) was retrieved from the Cross-Calibrated Multi-Platform wind vector analysis (CCMP) version 2, fitting radiometer and scatterometer data, in-situ observations, and ERA-Interim reanalysis into a 6-hourly, 0.25-degree resolution dataset (Wentz et al., 2015). Pointwise observations were retrived

with hourly frequency from 312 NOAA stations (NOAA, 2024) for sea-level pressure, near-surface temperature, and wind velocity, and from the Acqua Alta oceanographic tower (AA, 12°30.55′, 45°18.8′, see Figure 1a) for wind velocity.

**Table 2.** Overview of the observational datasets used for the validation.

| Name | Variables | Spatial coverage | Temporal coverage | Notes |
|---|---|---|---|---|
| EOBS | R, SLP, T | land, 0.1° | 1987-2010, daily | - |
| TRMM | R | land and sea, 0.25° | 1987-2010, daily | 3-hourly in the original dataset |
| CCMP | U, V, UV | land and sea, 0.25° | 1987-2010, daily | 6-hourly in the original dataset |
| NOAA | SLP, T, U, V, UV | pointwise, land | 1987-2010, daily | hourly in the original dataset |
| AA | U, V, UV | pointwise, sea (12°30.55′ E, 45°18.8′ N) | daily | hourly in the original dataset |
| TG | sea level | pointwise | variable | - |
| AVHRR | SST | 0.0417° | 1987-2010, daily | - |
| T2TS | $\theta$ | pointwise, 13°45′ E, 45°39′ | daily | - |
| CTD | $\theta$, S | pointwise profiles | variable | - |

Sea level information was provided by tide gauge (TG) datasets from 11 stations along the Adriatic coast (Table 3). The original time series consist of hourly sea level elevations. The data of Ancona, Marina di Ravenna, Vieste and Otranto come from ISPRA, Rome (Italian Institute for Environmental Protection and Research; https://www.mareografico.it); Venice data come from CPSM, Venice (Tide Monitoring and Forecast Centre; https://www.comune.venezia.it/node/6214); Trieste data come from CNR-ISMAR, Trieste (Raicich, 2023a, b); Bakar data come from the University of Zagreb (Medugorac et al., 2022, 2023) the data of Rovinj, Zadar, Split, and Dubrovnik come from HHI, Split (Croatian Hydrographic Institute). Sea levels at Rovinj, Bakar, Zadar, Split and Dubrovnik tide gauges were obtained by the mechanical instruments located in a stilling well, with 7-day charts digitized with the Auto-CAD software to obtain hourly sea level values.

Daily sea surface temperature (SST) observations at 0.0417° resolution have been retrieved from the L4 Optimal Interpolation (L4OI) Mediterranean Advanced Very High Resolution Radiometer (AVHRR) SST Analysis dataset (Pisano et al., 2016). A continuous multi-decadal time series for daily in-situ sea water temperature at 2-metre depth on the Trieste harbour station (T2TS, 13°45′ E, 45°39′) was adopted from Raicich and Colucci (2019). The observational reference for basin-wide characterisation of the thermohaline properties of the water masses was given by the CTD profile collection from different survey campaigns described by Pranić et al. (2021) and partially available at https://zenodo.org/record/5707773#.YmkprdpBxPY (Vilibić, 2021).

Due to the absence of the Ionian Sea dynamics in the model implementation and the consequent impossibility of properly reproducing the Adriatic-Ionian Bimodal Oscillating System (BiOS, see Civitarese et al. 2023) signal, the analysis of the sea level variability cannot replicate the approach by Pranić et al. (2021). Likewise, the absence of a very high resolution along the Croatian coast prevents the possibility of relying on the ADCP dataset used in that work for the validation of modelled circulation. Therefore, the scope of the assessment of these quantities in the present work is to verify that the dynamical properties of the basin are compatible with the observations and with the well-known basin-scale circulation features.

**Table 3.** Tide gauges from the TG dataset and their position in the model grid.

| Name | symbol | lat, lon (real) | lat, lon (mod) |
|---|---|---|---|
| Trieste | TS | 13.7595, 45.6473 | 13.7518, 45.6566 |
| Venice Punta Salute | VE | 12.3367, 45.4307 | 12.3451, 45.4206 |
| Bakar | BK | 14.5333, 45.3000 | 14.5080, 45.2819 |
| Rovinj | RO | 13.6283, 45.0833 | 13.6102, 45.0621 |
| Marina di Ravenna | RA | 12.2829, 44.4921 | 12.2957, 44.5104 |
| Zadar | ZD | 15.235, 44.1233 | 15.2330, 44.1480 |
| Ancona | AN | 13.5065, 43.6248 | 13.4885, 43.6443 |
| Split Gradska Luka | SP | 16.4417, 43.5067 | 16.4227, 43.4834 |
| Dubrovnik | DU | 18.0633, 42.6583 | 18.0514, 42.6154 |
| Vieste | VI | 16.1770, 41.8881 | 16.1748, 41.9019 |
| Otranto | OT | 18.4971, 40.1472 | 18.5272, 40.1375 |

The projection of oceanographic processes within a relatively small regional basin and in changing climate conditions is subject to different sources of uncertainty and different levels, from the evolution of the global climate to how this signal propagates through different scales and how the adopted numerical description impacts the final results. Extensively tackling all the possible sources of error requires the combination of different techniques and in principle an enormous effort, but the main elements of uncertainty can still be circumscribed at an affordable cost. First, evaluating the performance of the RCM-ROMS modelling chain by pursuing the "perfect boundary conditions" (Christensen et al., 1997) approach driving the RCM with a reanalysis allows to depurate the assessment from the intrinsic errors of the driving GCM. Nonetheless, since this kind of information is obviously not available for the future, the result of this operation is not automatically telling of the capability of the whole modelling chain (GCM-RCM-ROMS in this case) to actually reproduce the future climate. This aspect can be addressed by comparing modelled and observed statistics in the CTR period providing an aggregated description of the climate variability, under the assumption that the skills exhibited by a climate modelling system under reconstructed radiative forcings (as this is what ultimately drives GCMs in historical simulations) are representative of what can be obtained under projected conditions. Finally, the use of an ensemble approach can provide some degree of information about the uncertainty associated with different modelling strategies. The results of the evaluation (EV) run and of the atmospheric fields used as a forcing under "perfect boundary conditions" are presented and discussed in Section 3.1, whereas the overall climate model skills and uncertainty are addressed in Section 3.2, alongside with the projected future variations for some relevant quantities and processes.

# 3 Results and discussion

## 3.1 Evaluation

### 3.1.1 Atmospheric forcings

An overview of the skills of SMHI-RC4A forced by ERA-INTERIM reanalysis is given in Figure 3. Rainfall and wind appear as the most challenging variables to be properly reproduced, reflecting a recurrent behaviour in RCMs. Strongly spatially and temporally variable quantities like precipitation are intrinsically subject to large errors (Ban et al., 2021; Sangelantoni et al., 2023), and the basin orography and its effect on the land-sea-atmosphere interaction can add a strong element of complexity in the description of the process. Orographic control and the description of the land-sea transition are also a challenging element for the correct reproduction of wind fields, particularly in the Adriatic Sea (Signell et al., 2005; Bellafiore et al., 2012; Sanchez-Arcilla et al., 2021). Furthermore, although being the only available option for the evaluation of SMHI-RCA4, ERA-INTERIM does not presently represent the state of the art for atmospheric modelling, and is known to be far from the "perfect boundary conditions" hypothesis, particularly in terms of rainfall-related quantities (Bao and Zhang, 2013). In turn, slowly varying variables as sea-level pressure and near-surface temperature appear well reproduced also at the local scale.

Considering the spatial pattern of the mismatch between model and gridded datasets (Figure 4), the error on daily rainfall is on average mostly within $\pm 2$ mm, with a positive bias over the Apennines and western Balkan ridges and a negative bias over the sea. The largest discrepancies (1st and 99th percentiles) are mostly encompassed within $\pm 20$ mm d$^{-1}$ but can also concern the description of heavier precipitation events, possibly by more than 40 mm d$^{-1}$ on the southern Apennines and along the eastern coast and its mountain ridge. The bias on sea level pressure is within $\pm 3$ hPa in most of the domain, with a larger negative value in the southeastern part of the domain. A similar pattern can be found for 1st and 99th percentiles, with a minimum occurring in the same region as the only feature of an otherwise nearly uniform distribution. The near-surface temperature bias over land is negative (up to -3°C in mountain regions) throughout most of the domain, with smaller positive values in the northern coastal areas and in the far northeastern part of the mainland. Extreme values of the difference between model and observations are mostly bracketed in the $\pm 10$°C interval, with possible underestimates by more than $-12$°C in the inner mountain areas.

A separate consideration should be dedicated to the comparison between modelled and observed wind fields. Alongside with the documented tendency of CCMP to globally underestimate relatively strong ($>15$ m s$^{-1}$, see Mears et al. 2022) wind speed, the quality of this dataset in the Adriatic Sea is hampered by the resolution of the first-guess data source (ERA-INTERIM) and by the largely coastal setting of the basin, in which the use of satellite data for wind estimation is particularly challenging. To partially overcome this limitation, at least in the Northern Adriatic Sea, modelled winds are simultaneously compared with gridded CCMP data and in-situ observations at the Acqua Alta tower. Figure 5 thus shows a generally satisfactory model performance in reproducing the directional statistics of the wind regime for both the EV run and the climate ensemble, with particular reference to the dominance of northeasterly winds, although slightly overestimating the frequency of moderate to strong winds. In the meantime, while it is confirmed that CCMP underestimates the strong wind events, the directional

distribution suffers from a severe overestimate of the frequency of northerly and southwesterly wind, recalling the importance of adopting a specially critical approach when using this dataset for applications in this region.

The statistics displayed in Figure 4 are generally consistent, in terms of range and some features of the spatial patterns (e.g. minimum 1-percentile precipitation difference along the eastern coast, systematic SLP underestimation in the southeast) with those found by Denamiel et al. (2021a). Together with the skill metrics summarised in Figure 3 and wind statistics in Figure 5, this suggests that the quality of the atmospheric forcings used in the present study should not undergo major shortcomings with respect to a state-of-the-art kilometre-scale implementation and the wind regime is fairly well captured also in the challenging

coastal and orography-controlled setting of the Adriatic Sea. Besides, SMHI-RCA4 is known to show overall representative skills for essential climate variables as assessed in recent review articles including the large CORDEX ensemble (Coppola et al., 2021; Diez-Sierra et al., 2022; Vautard et al., 2021), and specifically over the Adriatic region where Belušić Vozila et al. (2019) consider wind climate specifically.

### 3.1.2   Sea level variability and circulation patterns

The evaluation of the model skills in terms of capability of reproducing sea level variability and hydrodynamics in this study follows a different approach from the analysis carried out by Pranić et al. (2021). In that work, sea level validation was mostly focused on the analysis of spatial EOF components and highlighted a relevant importance of the BiOS signal, which in the present study cannot be properly captured due to the exclusion of the Ionian Sea. In addition, most of the ADCP data used by Pranić et al. (2021) were available along the Croatian coast, where the model resolution can hardly be adequate to

reproduce circulation features largely controlled by local, and possibly complex, geometrical and bathymetric constraints. These considerations led to the choice of focusing instead on the pointwise comparison of modelled time series against tide gauge records and on the overall features of basin-scale circulation.

    Figure 6 presents the scatterplots of observed and modelled daily-averaged sea surface elevation data at 11 stations distributed along both sides, and at different latitudes, of the Adriatic coast. Worth recalling, all time series are depurated from the

linear trend and, in the case of observations, from the Sa and SSa tidal components. Although the model undergoes a systematic tendency to underestimate sea level variability within the basin, this limitation seems to progressively reduce for increasing latitudes, namely, for increasing distance from the southern open boundary condition. This is true also for diurnal and semidiurnal frequencies (not shown here), and the northbound improvement of the model skills suggests that this shortcoming could be due to an underestimate in sea surface level modulation in the boundary conditions, partially compensated by the internal

dynamics. This difficulty in reproducing tidal oscillations may contribute to some mismatch in circulation and tracer transport patterns over the short term, but since the result of the present validation is considered in aggregated terms, there seems to be no obvious reasons to consider this factor as a source of systematic errors, while it could likely contribute to add some noise in the average skill metrics. In any case, although the use of the present dataset for studies on sea level and its implications (e.g. coastal flooding) at the basin scale should probably require special attention (and most likely an intensive bias adjustment),

the fairly good performance on the Northern Adriatic coast permits a more straightforward use in this region, also in terms of boundary conditions for local downscaling applications.

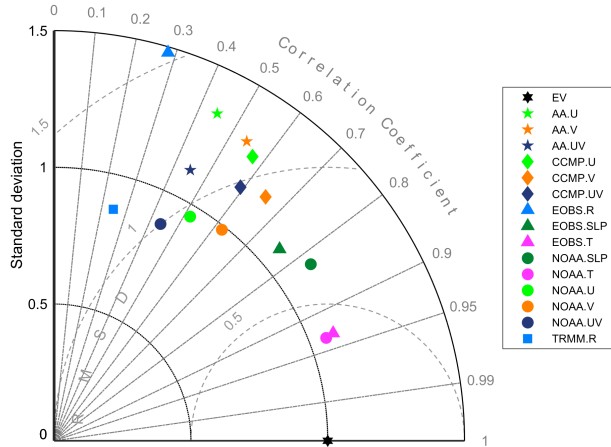

**Figure 3.** Taylor diagram summarising the normalised skills of the atmospheric forcings used in the evaluation run (EV) compared against the Acqua Alta (AA), CCMP, EOBS, NOAA and TRMM datasets.

The average modelled near-surface currents at the basin scale (a) and in the northernmost region (b), together with intermediate-depth patterns in the southern basin (c), is presented in Figure 7. Although the time-averaging smooths out the seasonal variability, the model appears capable to reproduce the main well-known circulation features (Lipizer et al. 2014 and references therein; Cushman-Roisin et al. 2001). Along the Eastern coast the inflowing Eastern Adriatic Current is well visible alongside with its cyclonic recirculations around the Southern Adriatic Pit (Southern Adriatic Gyre), the Jabuka Pit (Mid Adriatic Gyre), and the northern coast (Northern Adriatic Gyre), whereas the Western Adriatic Current flows southbound along the Italian coast. In the northern Adriatic, the signature of the typically wintry gyre encompassed between the Po Delta and the Istrian Peninsula (that is, north of 45°N) is partially visible also from the multi-decadal, year-round averaging, together with the cyclonic structure recirculating water masses from the northern Dalmatian islands to the Italian coast between the Po Delta and Ancona (43°36′ to 45°N, see Carniel et al. 2016). In the southern basin, in the 150-500 m depth range, the flow of the Eastern Intermediate Water is well visible along the continental slope and compatible with the climatological value of approximately $0.10 \, \text{m s}^{-1}$ (Orlić et al., 1992; Artegiani et al., 1997), suggesting that the good description of the thermohaline properties at the southern open boundary allows to recreate realistic geostrophically-controlled flow fields (as is the case for EIW) overcoming possible uncertainties in the 3D momentum boundary conditions.

### 3.1.3 Thermohaline properties

The assessment of the model skills in reproducing the thermohaline properties of the Adriatic Sea, as well as their variability over multiple time scales, is crucial for investigating its usability and possible limitations for a broad set of applications. Sea surface temperature (SST) is not only a key variable for the characterisation of air-sea heat fluxes and the numerical modelling

description of possibly intense meteo-oceanic events such as tropical-like cyclones and heavy precipitation events (Ricchi et al., 2017, 2021), but is also a reference parameter for the identification of extreme events such as Cold Spells (CSs) and Marine Heat Waves (MHWs), for which one of the most broadly accepted definition (Hobday et al., 2016) is based on the persistence respectively below the climatological 10th percentile or above the climatological 90th percentile for a period longer than 5 days. The difference between modelled (EV run) and observed (AVHRR) seasonal values for 10th, 50th and 90th percentiles are shown in Figure 8.

Overall, modelled SST statistics are generally characterised by relatively small errors, mostly bracketed in $\pm$ 1 °C and spatially distributed in patchy and mostly uncoherent structures. In winter (January to March), a cold bias affects the coastal regions of the western and south-eastern Adriatic, with an underestimate mostly smaller than -1 °C but particularly marked (exceeding -2 °C) for 10th percentile along the central Italian coast. In the same season, SST seems overestimated in the open

sea and in some segments of the northern and southwestern coast in a range between less than 0.5 °C (10th percentile) and 1.5 °C (90th percentile). In spring (April to June) the picture in the north appears reversed with a moderate overestimate (mostly < 1 °C) of 10th and 50th percentile SST and an underestimate (locally up to $-1.5$ °C off the Po delta) of the warmer conditions (90th percentile), though with the persistence of a small (generally < 0.5 °C) warm bias in the central and southern regions and a small cold bias along the southeastern coast. Summer (July to September) SST appear mostly subject to a variable

degree of underestimate, more marked (locally exceeding $-1$ °C) in most of the northern and central regions and along the southeastern coast, with a notable exception in the Kvarner Bay where a persistent positive bias (approximately 1 °C) occurs. In turn, modelled autumn (October to December) values appear generally overestimated in most of the basin, with a maximum exceeding 1.5 °C off the Po delta. In aggregated terms (Figure 9), the mismatch between modelled and observed SST values lies in the $\pm 3$ °C (respectively 1st and 99th percentile of the model-observations difference) in most of the basin, with larger

absolute values (locally up to $\pm 4$ °C) off the Po delta and, for 1st percentile, along the central Italian coast. This range is similar to the one discussed by Pranić et al. (2021) with reference to the AdriSC model, although the pattern in that work exhibited a more pronounced zonal gradient, with lower temperatures (that is, larger negative and smaller positive errors) along the western coast. Worth noting, also the modelled warming trends in the period (Figure 10), ranging between 0.25 and 0.40 °C per decade, are consistent with previous estimates (Mohamed et al., 2019; Amos et al., 2017; Tojčić et al., 2023) based on

different observational datasets and slightly different time periods.

     The analysis of the model results against the complete available three-dimensional thermohaline information can provide, alongside with a deeper insight on the model capability to capture the basin dynamics, a key for the interpretation of the SST-based skill assessment. In this direction, a first broad overview is given by Taylor diagrams and quantile-quantile plots referred to the CTD dataset in the subdomains, including the surroundings of the Otranto Strait, and in the whole Adriatic Sea

(Figure 11). Model skills in terms of potential temperature ($\theta$) appear generally better, and less variable among subdomains, than in terms of salinity (S). Overall, intermediate values of $\theta$ tend to be overestimated by the model, particularly in the Northern Adriatic subdomain, whereas the tails of the potential temperature distribution are generally well reproduced in all subdomains. In turn, while intermediate to high S values are mostly well reproduced, low to mid salinity tends to be overestimated, particularly in some subdomains. In general, uncertainties in freshwater discharge represent a main source of

possible error for salinity: while in NA and WC the overall contribution is larger but given mostly by surface runoff, and therefore more controllable and easier to quantify, in the northeastern and eastern coast a large fraction of the freshwater supply comes as karstic groundwater sources, whose quantification is recognisedly challenging and subject to potentially large error, particularly at KB and DI. Furthermore, the model resolution does not permit a complete description of the complex geomorphology of that coast, leading to some shortcomings for small-scale circulation features and, as a consequence, on
tracers (such as salinity) patterns. Like in the case of SST, also skills for 3D $\theta$ and S are comparable with the values found for AdriSC by Pranić et al. (2021), although the metrics discussed in that work consider all the subdomains aggregately while distinguishing among different campaign datasets. Importantly, skill metrics are consistently good also in the OS region, providing some confidence on the quality of the boundary conditions. In terms of potential density anomaly ($\sigma_\theta$), intermediate values are generally underestimated (in the Northern Adriatic, which is the most unfavourable situation, up to approximately
0.6 kg m$^{-3}$), but for higher values the performance tends to improve, with the mismatch progressively decreasing to less than 0.2 kg m$^{-3}$ for $\sigma_\theta$ greater than 29.4 kg m$^{-3}$. Importantly, the comparison between EV and EV$^*$ runs in the Taylor diagrams also show a very small influence of the version of the CMEMS product used for the boundary conditions on the overall statistic. If the use of the latest version of MFS (Escudier et al., 2020), released after the implementation of the climate ensemble, for the EV$^*$ run does not significantly improve the model skills, it is reasonable to expect that the use of a previous dataset (Simoncelli
et al. 2019, the only one available at the time of the ensemble setup) to compute the climatologies at the boundary is not a possible important source of shortcomings in the climate runs. Although again aggregated at the subdomain scale, Figure 12 shows how model errors are distributed over time and along the water column. Here we focus on the Northern Adriatic (panels a and d), Kvarner Bay (b and e) and Deep Adriatic (c and f) due to the relevance of these areas for dense water dynamics and the contribution of the Adriatic Sea for the Mediterranean thermohaline circulation.

In the Northern Adriatic, $\theta$ medians exhibit a very close match between model and observations during the winter months, while the temperature overestimate progressively increases in spring and summer, and mostly below the 15-metre depth, likely as an effect of excess of vertical mixing. Such a heat content surplus is then redistributed throughout the water column in autumn and progressively reduced to very small values. The seasonal variability is not as clear for $S$ in this region, but in this case the vertical distribution of the error reaches its maximum values close to the surface, possibly reverberating the uncertainty
in the description of freshwater inputs and its implications for plume dynamics. The performance in terms of $\sigma_\theta$ appears mostly controlled by temperature values, with larger errors for higher depths and better agreement in winter and close to the surface. In the Kvarner Bay the overestimate in summer temperature, with a similar profile as in the Northern Adriatic, is particularly evident compared with the other seasons, while $S$ remains slightly overestimated throughout the whole year, with larger errors concentrated in the upper 10 metres. In this case, near-surface $\sigma_\theta$ errors appear mostly controlled by salinity, particularly in
spring and summer, whereas $\theta$ errors control the winter profile of the error and the sub-surface part of the summer profile. In the deep Adriatic, a moderate underestimate of the depth-averaged climatological values of $\theta$ and $S$ throughout the year reflects a consistent pattern along the intermediate-depth regions of the water column, that is, 200-900 m for $\theta$ and 200-600 m for $S$. The overestimate in $\theta$ in the sub-surface layer (approximately between 10 and 100 m depth) reported in summer and autumn reflects the patterns observed in the northern regions, and is compatible with a possible excess in vertical mixing in the

upper layers. By contrast, the observed underestimate of both quantities for larger depths could be inherited from the dataset used for initialization and boundary conditions: in fact, the results presented by Pranić et al. (2021) based on the same dataset showed very similar values, and the data from the OS subdomain (although based on an insufficient number of observations for a robust spatial and temporal breakdown, and therefore not shown here) tend to qualitatively confirm the same pattern in the near-boundary region. A conclusive interpretation of the origin of this mismatch would require a dedicated effort, but in any case the evidences presented here allow to consider that the main features of the water masses in this region are properly represented (see also Figure 2), particularly and most importantly in the deeper layer, which is the one where deep ventilation occurs and whose $\sigma_\theta$ background values exert a fundamental control in dense shelf water downflow.

### 3.1.4 Extreme thermal events

Extreme events are typically an element of major interest in climate projections. With specific reference to thermal extremes, besides the well-acknowledged role of wintry cold air outbreaks in dense water formation in the Adriatic Sea, there is increasing awareness of the potential effect of Marine Heat Waves (MHWs) and Cold Spells (CSs) on marine systems, with particular concern on the impacts on coastal and transitional environments (see for the Northern Adriatic Sea Ferrarin et al. 2023, and references therein). In this perspective, a separate section of the evaluation is dedicated to the assessment of the EV run to capture the key features of observed extreme events. MHW (CS) are identified following Hobday et al. (2016), as periods longer than 5 days in which sea surface temperature is persistently above (below) the daily climatological 90th (10th) percentile in the reference period (an 11-day sliding window was considered in this application), while the cumulative intensity is defined as the difference between current value and climatological daily mean integrated over the event duration. Focusing on the in-situ records collected at Trieste, Figure 13 shows that timing (panel c) and intensity (panel d) of these events are generally well reproduced. In particular, the modelling chain seems to satisfactorily capture most of the features of the interannual variability and the alternation of ordinary periods (e.g. 1995-1998) and exceptional years (e.g. 1987, 2001, 2007), although with some remarkable exceptions such as in 2006. The intra-annual variability of the occurrence of thermal extremes appears also at least partially well captured, with some increase in MHW occurrence in summer (worth recalling, this is not obvious, as the definition of these events in different periods throughout the year is referred to the statistics of the same period) occurring also in the EV results (panel e), while some shortcomings appear in the reproduction of the very weak modulation of CS occurrence (panel f). A similar assessment (not shown here, but available in Bonaldo et al. 2024b) carried out for each subdomain using the SST from the AVHRR dataset (Pisano et al., 2016) confirms that the results shown for Trieste are actually representative for the performance in the whole domain.

### 3.2 Climate historical runs and projected climate change signal

If the focus on the EV run allows to investigate the model skills in the presence of the best available, if not actually "perfect", information, the analysis of the climate ensemble in the recent past provides, besides of course a terms of reference for comparison against the future figures, a further element to complement the evaluation by verifying to what extent the observed climate statistics in the recent past are well reproduced also in a GCM-driven condition. In this direction, and again with reference to

the three subdomains considered in Figure 12, the climate normals of the monthly values of 1st, 50th and 99th percentile of SST in different data sets are shown in Figure 14 including also statistics from observed data and the EV run. As a recurring pattern, the climate ensemble statistics appear satisfactorily matching the observations in the winter months, while underestimating and overestimating respectively summer and autumn values, most likely as a result of the description of the fluxes of heat along the upper layers of the water column associated with the excess mixing described in Section 3.1. This behaviour is more evident in the Northern Adriatic, where the maximum mismatch reaches 2.7 °C for the 99th percentile in June. The ensemble spread is generally narrow (around 1 °C) from October to April both in CTR and SCE conditions, and significantly larger (> 4 °C) in summer, particularly in the SCE datasets, in some cases obscuring the statistical significance of the future change signal. An examination of uncertainty partitioning through the different modelling chain steps lies beyond the scope of the present study, however, speculations can be made about the well-known large uncertainty charactering GCMs in reproducing crucial mid-latitude summer season dynamics like blocking atmospheric patterns (Davini and D'Andrea, 2020) and their response to a warmer climate (Woollings, 2010; Woollings et al., 2018). Nevertheless, also a local scale forcing can be expected behind resulting ensemble variability and exerted by the nested simulations (SMHI-RCA4 and ROMS) given the non-linear ingestion of the GCM large-scale signal.

The SST increase between SCE and CTR is generally evenly distributed throughout the year and among the different statistics, ranging in most cases, for ensemble means, between +2.8 and +3.2 °C, except for higher percentiles in summer, in which the increase ranges between +3.2 and +3.8 °C in most of the basin (Figure 15). While spatial patterns are relatively uniform in spring and summer, north-south and coast-offshore gradients are visible in autumn and winter, in agreement with the patterns in seasonal temperature change shown for the first time in the Mediterranean region by Giorgi and Coppola (2007). Locally higher values appear for 10th percentile SST along the southwestern coast in winter, while conversely warming seems to be generally milder in the Northern Adriatic. Sea surface salinity (SSS, Figure 16) is projected to increase, at different rates, throughout the basin with the only major exception of the northeastern coast in winter, in which very small or statistically non-significant variations are envisaged as a consequence of the increase in river runoff in this season (Bonaldo et al., 2023) counteracting the generalised salinisation trend. Conversely, in the same region and more broadly in the northern and western basin the highest increase in $S$ is projected in spring and even more in summer, when river runoff is expected to decrease. The picture is less clear for net surface heat fluxes (Figure 17), in which a tendency to some decrease (up to -40 W m$^{-2}$, although with an increase of summer heating in the north) is predominant on patchy patterns characterised by a north-south gradient throughout the basin.

Specific applications can benefit from a thorough comparison of the cumulative distributions of SST in the CTR and SCE periods. In particular, with reference to thermal extremes, any difference in the variations of the 90th and 99th percentiles, or of the 1st and 10th percentiles, is associated with a change in the shape of the tails of the statistical distributions for the two periods, and could be a possible source of variability respectively in MHW and CS statistics. Figure 18 represents the daily SST cumulative distributions for the ensemble (again considering an 11-day sliding window) in different sub-basins. For low-temperature extremes, narrower distribution tails can be found in winter in NA and KB (0.5 °C), and, to a lesser extent WC (0.3 °C) and MA (0.2 °C), and slightly broader in late spring and early summer (NA, KB, MA) and autumn (0.3 to 0.5

°C, all subdomains). Conversely, for higher temperatures projections show moderately narrower distributions in late summer in NA, MA, and WC (0.2 to 0.4 °C), and broader tails in late winter (0.3 to 0.5 °C) and autumn (0.2 to 0.4 °C) throughout the whole domain. A more comprehensive overview on how climate change affects the statistics of the thermohaline properties of the Adriatic Sea (and of the subdomains considered in this study) along the whole water column can be drawn from the quantile-quantile plots depicted in Figure 19.

While the projected potential temperature increase in the colder (and deeper) regions of the deep Adriatic is confined below 1.5 °C, the statistics throughout the different basins reflect the pattern presented in Figure 14, with variations mostly clustered around +3 °C. Projected ensemble average salinity increase is mostly encompassed between 0.3 and 0.4, with larger variations on the higher end of the distribution (namely, for S≥39). In terms of $\sigma_\theta$ variations, this results in a generalised tendency towards a decrease between 0.4 and 0.5 kg m$^{-3}$ in most of the basin (with the larger decrease corresponding to smaller values), with the exception of the deeper regions of the deep Adriatic, presently characterised by $\sigma_\theta$ around 29.2 kg m$^{-3}$ facing a decrease of approximately 0.2 kg m$^{-3}$. Most notably, Figure 19 shows that the variability of the results within the ensemble is generally larger than the variability across the subdomains: since the evidences from the EV run (Section 3.1) support a good degree of confidence in the model capability of reproducing the internal dynamics of the Adriatic Sea, this result gives an important account of the relative weight of the GCM-RCM modelling chain in the ocean climate projections at the basin scale. Figure 20 summarises the seasonal modulation of the variation of thermohaline (median) quantities along the water column, again with a focus on the relevant basins for dense water formation and deep ventilation. NA and KB are characterised by similar results, in both qualitative and quantitative terms. In NA (KB), the ensemble-mean increase in median $\theta$ values ranges between +2.5 (+2.6) and +2.7 (+2.8) °C in winter and spring, and between +2.9 (+2.9) and +3.1 (+3.2) °C in summer, with intermediate variations between +2.7 and +2.8 (+2.8, nearly uniform) °C in autumn. $S$ exhibits a more pronounced vertical variability. For depth smaller than 40 m, increases range between +0.28 (+0.37) in winter and +0.71 (+0.49) in summer, whereas for higher depths the range of the increase lies between +0.34 and +0.43 (+0.32 and +0.43). Median $\sigma_\theta$ is thus projected to decrease by -0.27 (-0.29) kg m$^{-3}$ in winter, when dense water formation typically takes place, and between -0.27 (-0.36) and -0.42 (-0.40) kg m$^{-3}$, with the largest values for depths larger than 40 m, in autumn. Variations in spring and summer, respectively in the range -0.34 (-0.36) to -0.32 (-0.34) and -0.55 (-0.57) to -0.41 (-0.46) kg m$^{-3}$, although with relevant values, are less significant for the thermohaline circulation in the basin. In DA, the seasonal modulation of climate change on median profiles is mostly visible for h≤200 m; for larger depths, thermohaline quantities vary gradually, and with negligible inter-seasonal differences, up to uniform values for h≥800 m. In the upper layer, variations range between +2.5 and +3.2 °C for $\theta$ and between +0.26 and +0.37 for $S$, again with smaller variations in winter and spring and larger variations in summer and autumn. $\sigma_\theta$ variations range between -0.40 kg m$^{-3}$ in spring and -0.71 kg m$^{-3}$ in summer. Below the upper layer, $\theta$ increase varies from + 2.8°C for h=200 m to +1.3°C for h≥800 m, $S$ increase varies from +0.21 to +0.17, and $\sigma_\theta$ varies between -0.44 and -0.15 kg m$^{-3}$. Also considered in the light of Figure 12, these results indicate that the Adriatic Sea should remain a cooling pool for the Mediterranean Sea, whereas its characteristic behaviour as a dilution basin is expected to be significantly reduced, particularly in summer when the highest salinity increase is expected. In fact, the rate of increase prescribed as a boundary condition (see

Figure 2), and consistent with other studies (e.g. Parras-Berrocal et al. 2020 for future projections), is not sufficient to explain such comparatively higher values, which appear mostly controlled by the decrease in summerly river runoff.

Before focusing on thermal extremes such as MHWs and CSs, it is worth recalling that the definition of these events (Hobday et al., 2016) is intrinsically associated with some definition of impact, in most cases in the framework of the discourse on climate change. In this direction, the choice of the baseline period as a reference for the computation of the threshold percentiles implies an important assumption on the system on which MHWs and CSs are supposed to act as stressors. More precisely, defining these events in a future scenario with reference to a past climatology implicitly assumes that the target system has limited resilience to warmer conditions (as could be the typical case for human civilizations). In turn, defining these events with reference to a future climatology is compatible with the assumption that the system can adapt to the change in the ordinary conditions and is only (or mostly) vulnerable to significant deviations to those conditions. In the present study, taking as reference thresholds climatological values from the CTR period yields the simple, though important, result that end-of-century conditions under RCP8.5 climate scenario are persistently corresponding to MHW. This is consistent with the evidence from the recent past, in which the increase of MHW in the Adriatic Sea and Eastern Mediterranean has been estimated as high as 100 days per year in the 1982-2020 period (Juza et al., 2022). The other hypothesis is considered in the plots in Figure 21, in which MHWs and CSs are defined for CTR and SCE, as well as for the EV run and for observations in the control period, with reference to the climatologies for the corresponding periods. Under this approach, modelled differences between SCE and CTR conditions (expressed as monthly mean cumulative intensity of the events) appear generally minor and in any case only occasionally statistically significant. These slight variations appear qualitatively consistent with the changes in the statistical distributions presented in Figure 18. The weaker correlation in summer months suggests that the variation in that period could be also associated with the duration of the events rather than with their maximum intensity. In any case, this would suggest that climate change impacts on ecosystems could be mainly controlled by the warming trend of the ordinary conditions, with only a secondary contribution from the change in the characteristics of extreme events. Nonetheless, it is worth noting that the RCM-ROMS modelling chain exhibits some shortcomings in properly capturing the cumulative intensity of extreme events between late spring and early summer (mostly in May and June). While the mismatch is generally fairly small in the case of the EV run (although with the caveat that the reference period is slightly different), this is more evident in the case of the CTR run, thus revealing, alongside with the limitations already pointed out in the model performance (e.g. the excess in vertical mixing), some limitations in the GCM-RCM capability to reproduce the extremes during the warm season. Importantly, there is an increasing evidence of a generalised tendency of GCMs and RCMs to underestimate temperature extremes historical trends, mostly due to the neglection of aerosol changes (see for instance Schumacher et al. 2024), therefore this specific result should be treated with some caution.

## 4 Conclusions

The present paper introduces a six-member kilometre-scale ocean model ensemble tackling end-of-century changes in the dynamics of the Adriatic Sea under a RCP8.5 climate scenario. Up to our best knowledge, this is the first effort undertaken to

characterise the effects of climate change on ocean dynamics in this region by combining the detail of the high resolution and a measure of the uncertainty as provided by the ensemble approach. The aim of this work is to pave the way for an extensive variety of multidisciplinary studies related to climate impacts on the Adriatic Sea, ranging from the possible changes in deep sea ventilation regimes to the dynamics and evolution of marine habitats, also including downscaling for local applications in coastal and transitional systems. In this direction, special attention was dedicated to a thorough assessment of the model skills, whereas the climate projections have been introduced in terms of expected variation and uncertainty on thermohaline quantities, with a focus on extreme thermal events. The set of processes and statistics addressed in the validation is meant to give an overview on the applicability and limitations of the model ensemble and provide some guidance to a broad range of applications. The resolution of the atmospheric model (0.11 °) might in principle be too coarse to properly reproduce extreme events, like the bora wind, bora-driven ocean circulation and formation of dense water (Kuzmić et al., 2015; Denamiel et al., 2021b; Pranić et al., 2023). This particularly applies to the complex coastal basin of the Kvarner Bay over which the bora-driven heat uptake reaches its maximum (Janeković et al., 2014) and which is recently assessed to contribute about 25-35% to the overall dense waters (Mihanović et al., 2018). Also, the cascading of dense waters in the Southern Adriatic Pit might be underestimated, as the model resolution in this area is probably too coarse to properly capture submarine canyons in which the dense waters are known to cascade (e.g., Paladini de Mendoza et al. 2022). Further, AdriE ensemble models have no capacity to address BiOS-driven quasi-decadal variability in thermohaline properties of the Adriatic, as requiring the inclusion of the northern Ionian Sea (in which the BiOS-driven circulation regimes) into the domain (Denamiel et al., 2022). Nonetheless, as confirmed by the comparison against the previous work by Pranić et al. (2021), purportedly carried out for the evaluation run wherever possible, shows that the performance of the SMHI-RCA4 - ROMS modelling chain is mostly aligned with the skills of a state-of-the-art kilometre-scale hindcast, in particular over the northern Adriatic where the resolution of the ocean model is at the kilometre-scale (ca. 2 km). In any case, for studies focused on specific processes (e.g. plume dynamics, lagrangian transport, fluxes across the continental margin) a dedicated validation is strongly recommended. Those applications will also provide a sound opportunity to explore the role and the relative weight of climate variations in atmospheric dynamics, Mediterranean-scale ocean properties and hydrological regimes, which is certainly recommended for individual processes but whose discussion in general terms is beyond the scope of the present study, and probably too broad for a single paper. The general scope of the dataset presented in this work also led to the decision to focus the discussion on the raw model outputs, while the possible bias adjustment strategies should be decided from time to time for each specific application based on its characteristics and on the trade-offs involved (Enayati et al., 2021).

Concerning the climate projections, the main results presented in this work can be summarised as follows:

- in ensemble-average terms, end-of-century projected SST increase is encompassed between +2.8 °C and +3.2 °C, with an uncertainty range of approximately 1 °C and in winter and up to 4 °C in summer;

- in general, the variation in thermohaline quantities is also larger, in absolute terms, in summer and autumn and smaller in winter and spring;

- over the considered time span, the variability of the change in thermohaline quantities within the ensemble is larger than across the subdomains, suggesting that any additional detail in the long-term projection deriving from a kilometre-scale approach could be curbed by the uncertainty in the regional and global climate evolution, if these are not properly taken into account;

- with reference to the recent past statistics, future conditions could be assimilated to a massive, persistent marine heat wave; conversely, taking as reference the future "ordinary" conditions (that is, implicitly assuming that the target system, however defined in socio-ecological terms, has adapted to the new state), the model ensemble does not provide strong evidences of major variations in the statistics of the thermal extremes. Worth noting, the observed shortcomings in the climate modelling chain capability to reproduce thermal extremes in summer suggest that this result should be taken with special care.

Monthly-averaged fields for the main oceanographic quantities from the climate simulations, as well as fields from the EV run, are publicly available on Zenodo (Bonaldo et al., 2024a), whereas specific requests for other variables or time resolution can be submitted to the corresponding author.

*Data availability.* Monthly averages of the main quantities are available on Zenodo (https://zenodo.org/records/11202265), and subsets of the full modelling dataset can be requested to the corresponding author.

*Author contributions.* DB coordinated the study and the preparation of the manuscript, DB, AR, and LS set up the model runs, DB, CD, PP, FR, AR, LS, and IV contributed in the model validation and analysis, all authors participated in the discussion of the results and in the editing of the text.

*Competing interests.* The authors have no competing interests to be declared

*Acknowledgements.* We acknowledge the E-OBS dataset from the EU-FP6 project UERRA (https://www.uerra.eu) and the Copernicus Climate Change Service, and the data providers in the ECA&D project (https://www.ecad.eu). The contribution of all the organizations that provided the CTD observational data used in this study is also gratefully acknowledged. The work of IV is supported by the Croatian Science Foundation projects C3PO (Grant IP-2022-10-9139) and GLOMETS (Grant IP-2022-10-3064), and Interreg IT-HR project AdriaClimPlus.

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

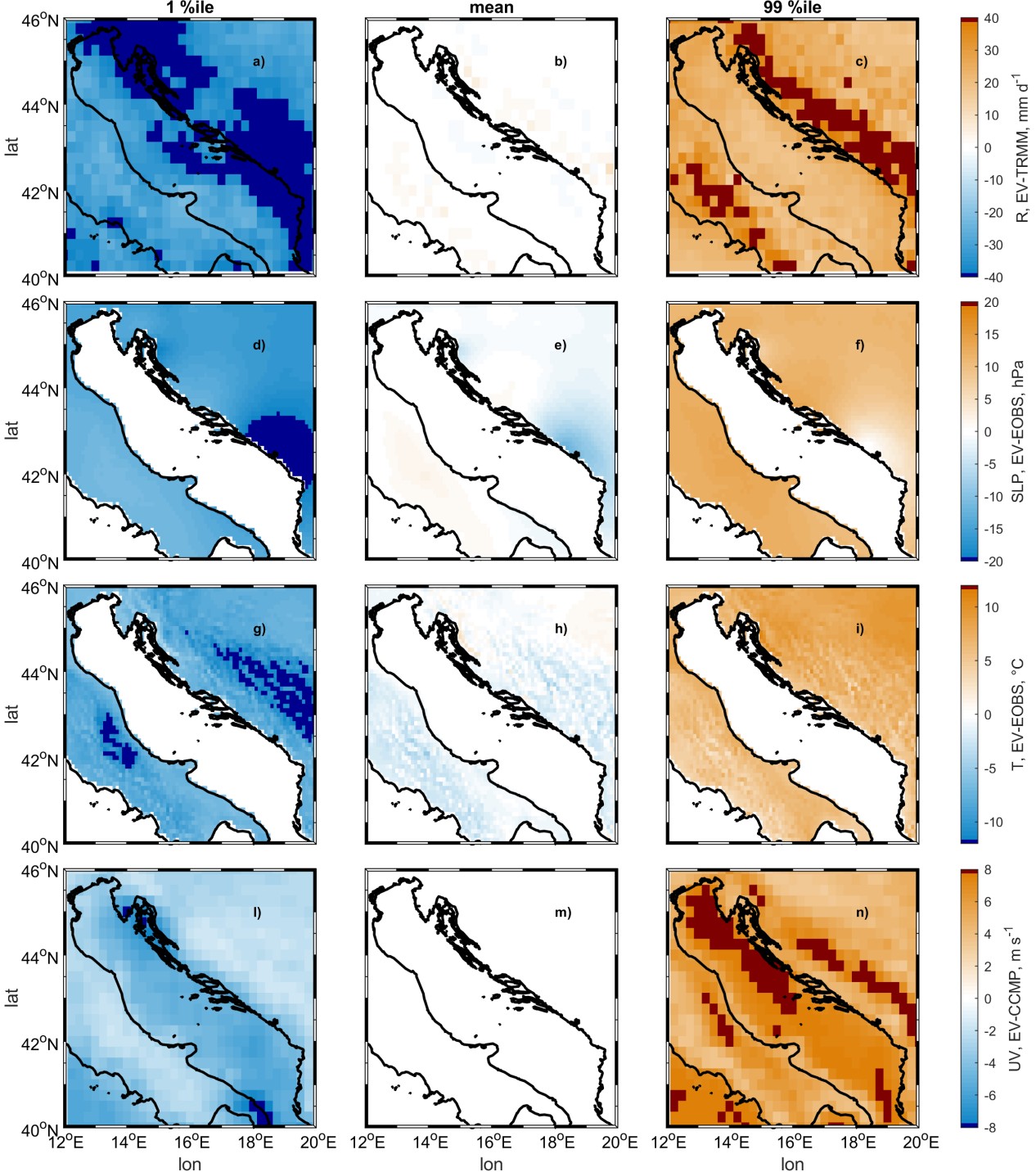

**Figure 4.** 1- percentile (a,d,g,l), mean (b,e,h,m), and 99- (c,f,i,n) percentile of the difference (*Bias* in Denamiel et al. 2021a) between atmospheric forcing in EV run and different gridded observational datasets.

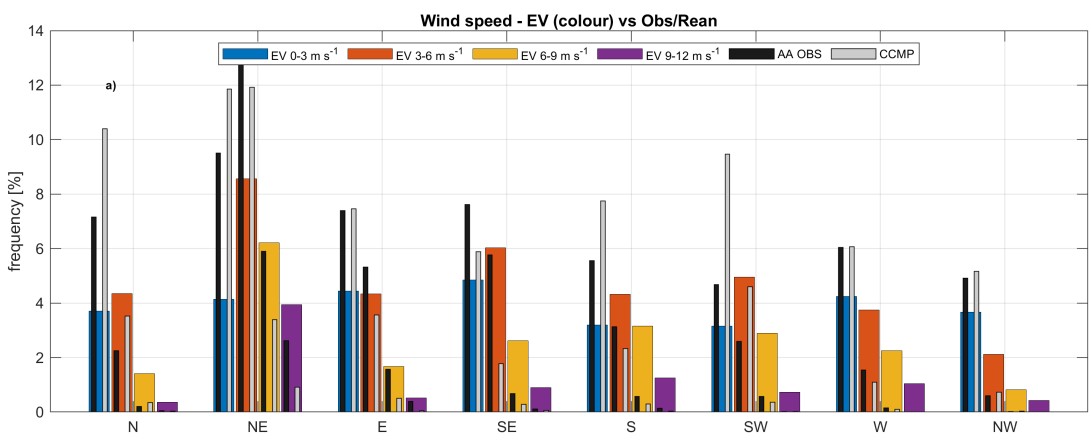

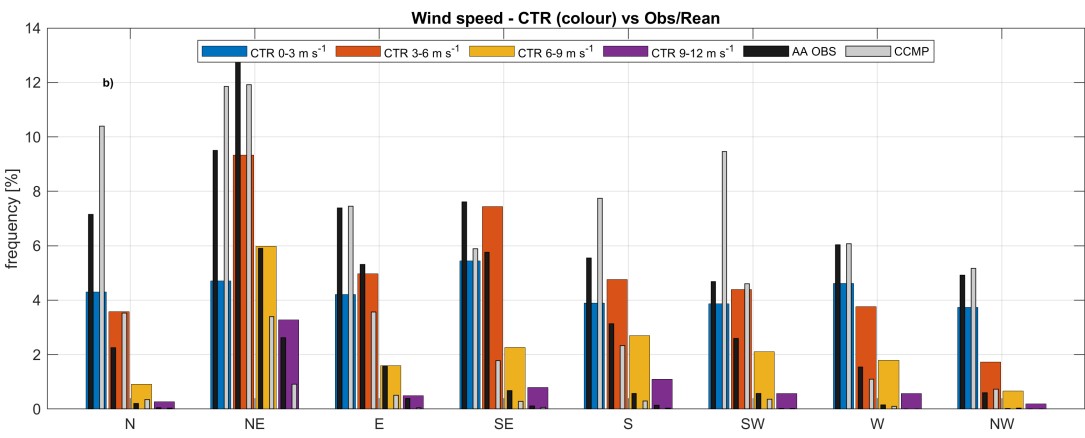

**Figure 5.** Comparison of SMHI-RCA4 fields used: a) in the EV run, and b) in the climate ensemble (coloured bars) against CCMP directional wind statistics and in situ observations (gray and black bars respectively) at AA in the reference period (July 10, 1987 - December 31, 2010).

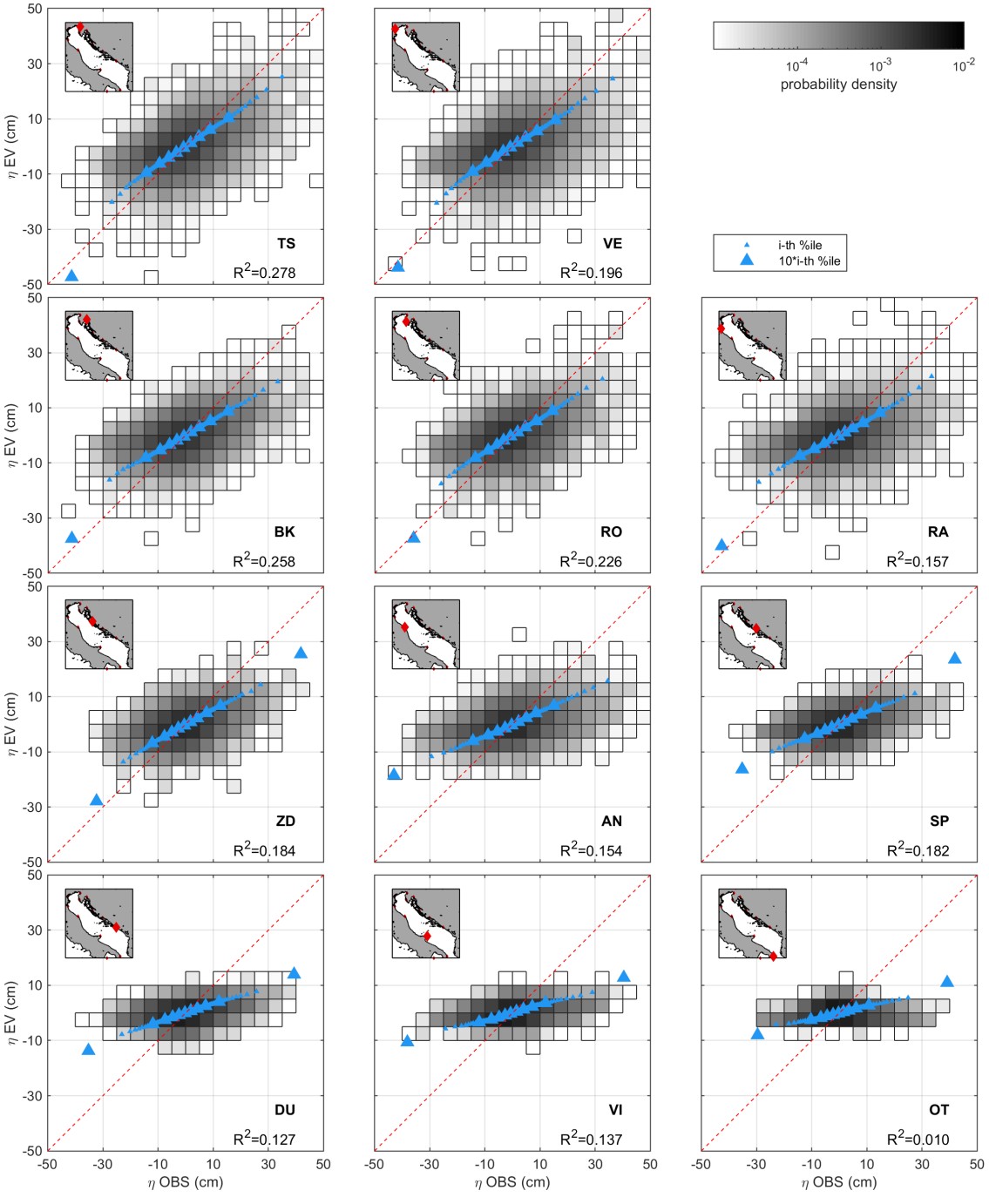

**Figure 6.** Scatterplots of modelled (EV) and observed (OBS) sea levels at different tide gauges and their percentiles. $R^2$ represents respectively the correlation coefficient for the whole series.

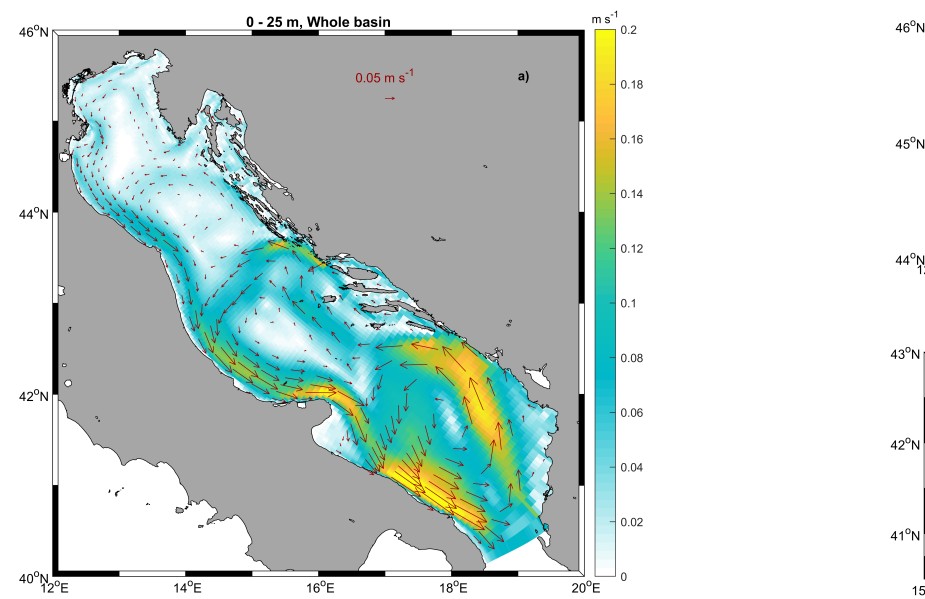
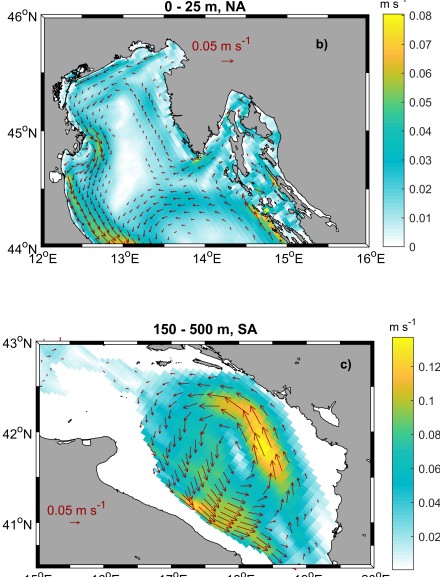

**Figure 7.** Mean near-surface (0-25 m) circulation patterns in the whole basin (a) and in the Northern Adriatic Sea (b), and at intermediate-depth (150-500 m) un the Southern Adriatic Sea (c) in the EV run. Vectors have been subsampled every 5 (a) and 3 (b,c) grid points, omitting values smaller than 0.01 m s$^{-}$1.

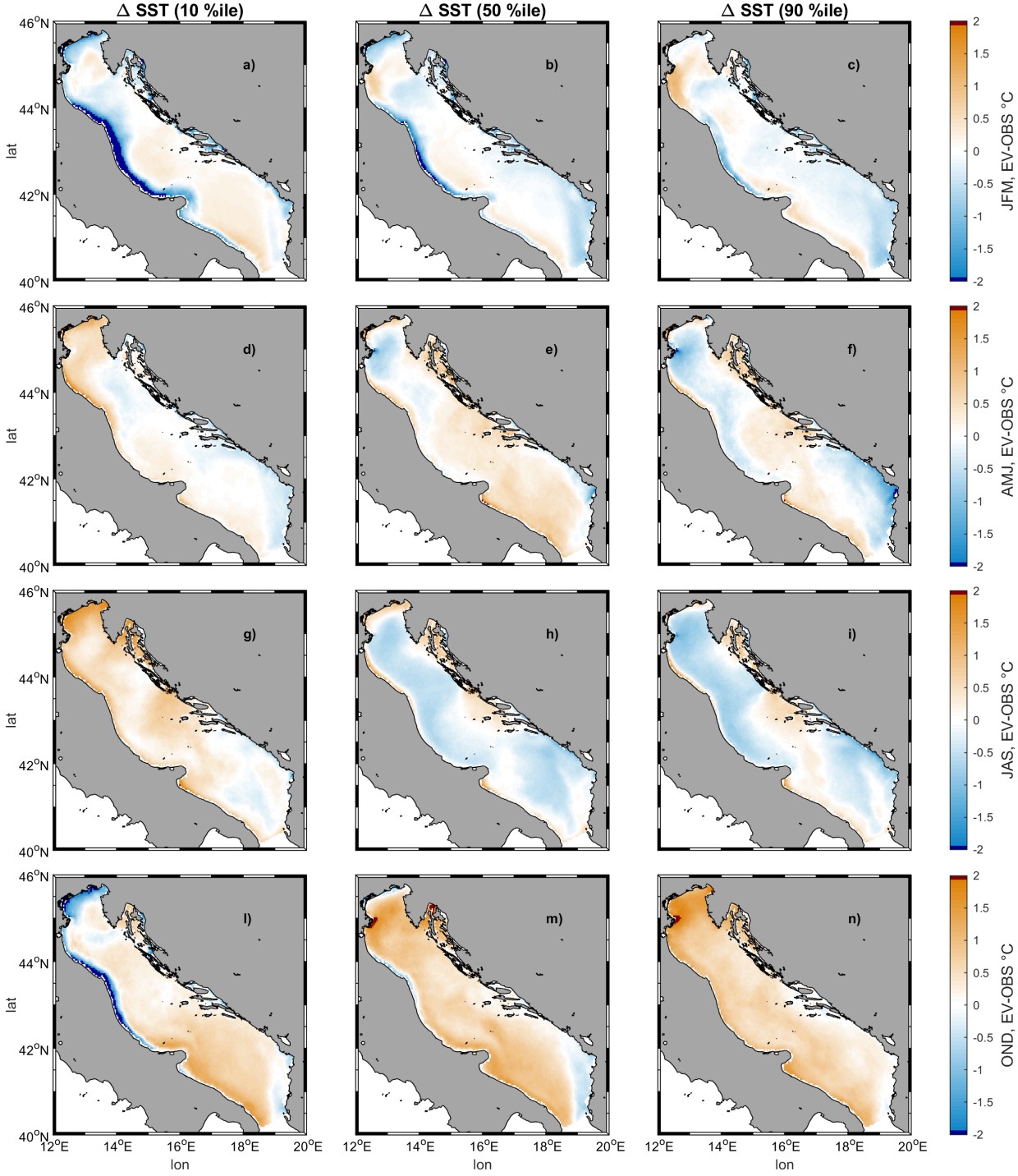

**Figure 8.** Patterns of difference between modelled (EV run) and observed (AVHRR) seasonal SST percentiles.

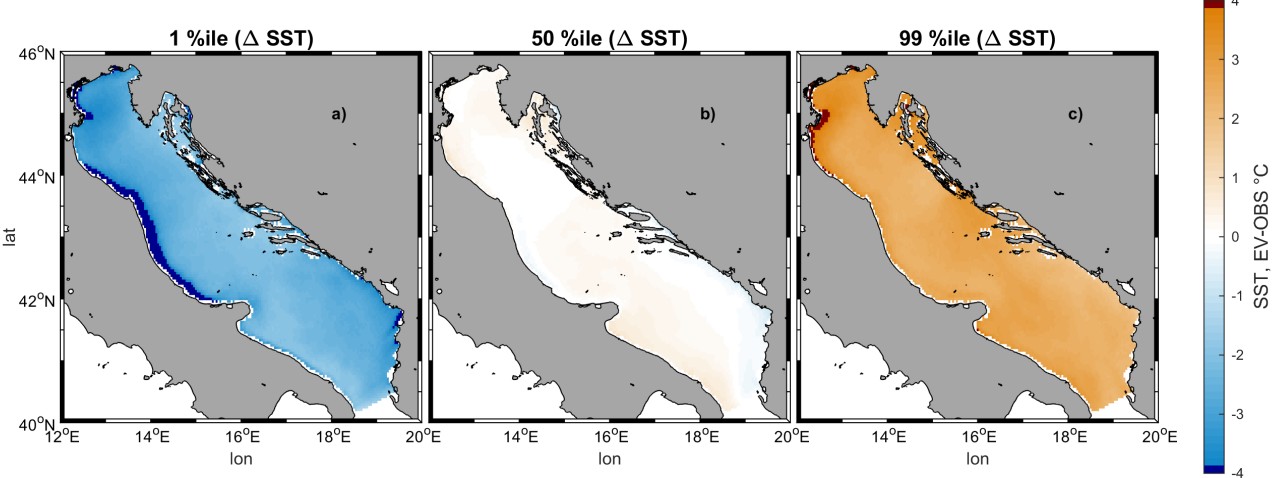

**Figure 9.** Percentiles (1st, 50th, and 99th) of difference between modelled (EV run) and observed seasonal SST.

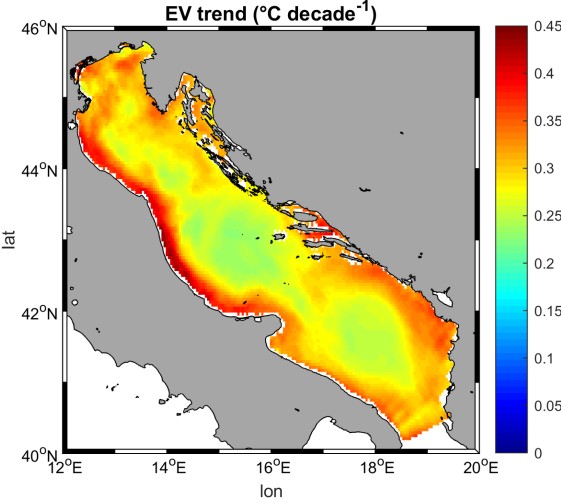

**Figure 10.** Modelled (EV run) SST trends in the basin, reference period 1987-2010. All values are statistically significant following a Mann-Kendall test.

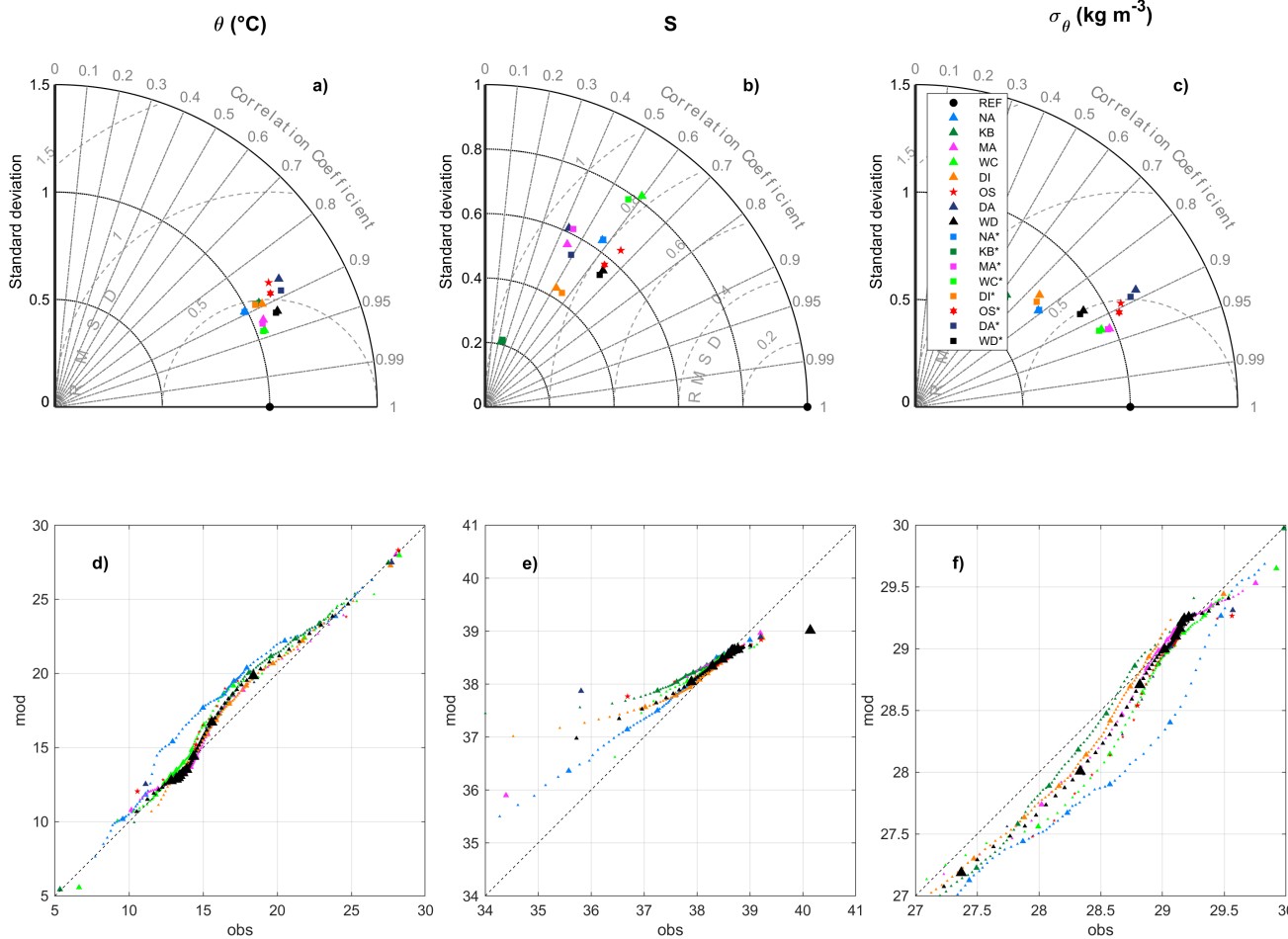

**Figure 11.** Overview of the EV run performance in reproducing the properties of the water structure, namely potential temperature ($\theta$), salinity ($S$), and potential density anomaly ($\sigma_\theta$). All plots refer to the subdomains identified in Figure 1b, i.e. Northern Adriatic (NA), Kvarner Bay (KB), Middle Adriatic (MA), Western Coast (WC), Dalmatian Islands (DI), Otranto Strait (OS), and Deep Adriatic (DA); WB represents the Whole Basin. Panels a-c: Taylor diagrams for the EV and EV∗ runs; Panels d-f: Q-Q plots for the EV run, with small markers every quantile and larger markers every 10 quantiles.

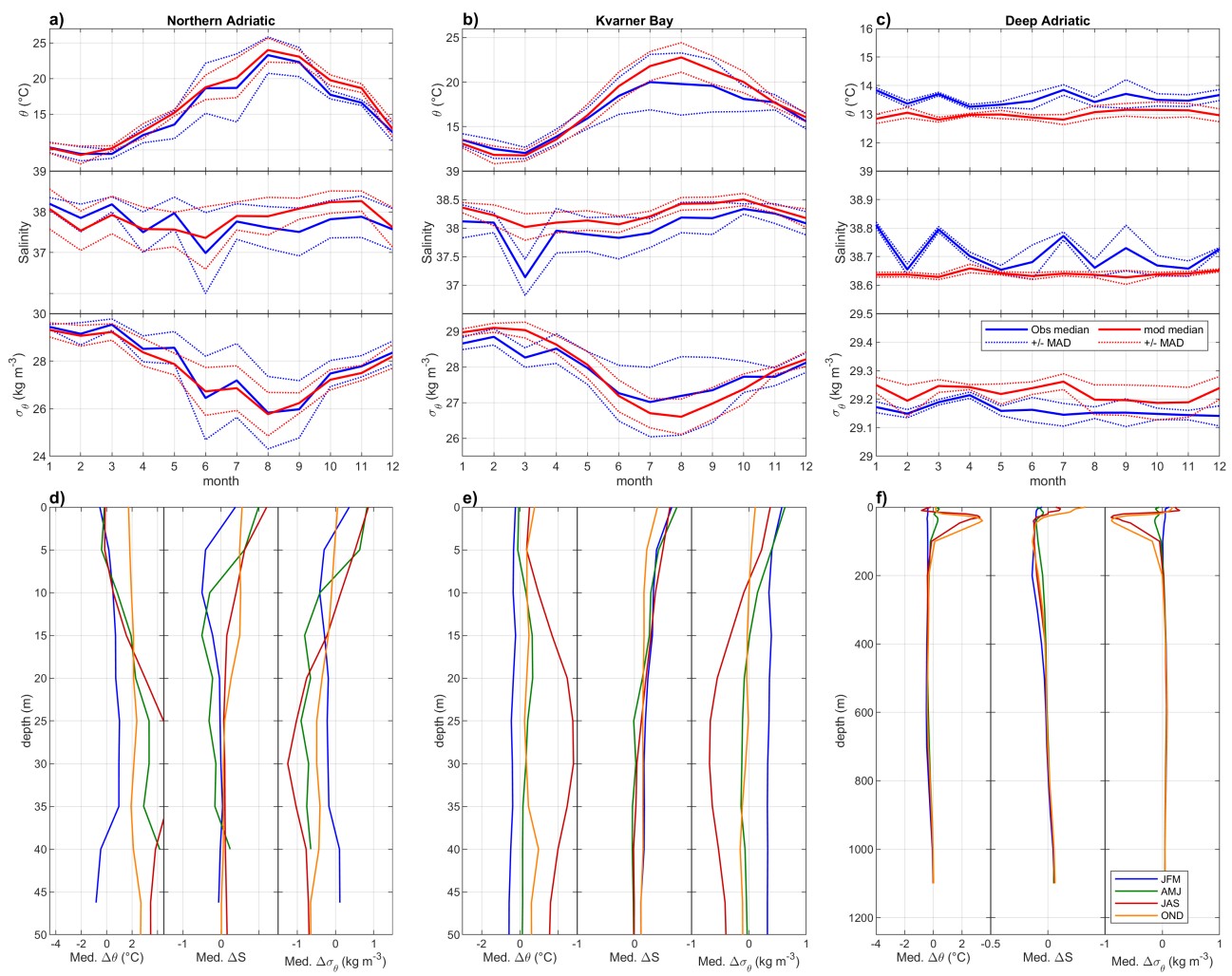

**Figure 12.** Monthy climatologies of modelled (EV run) and observed (CTD dataset) median potential temperature, salinity and potential density anomaly for the Northern Adriatic (a), Kvarner Bay (b), and Deep Adriatic (c). Dotted lines represent the median values ± the mean absolute deviation (MAD) for the dataset. For the same subdomains, seasonal profiles (in this case seasons have been defined as in Pranić et al. 2021 in order to facilitate the comparison) of median mismatch between model and observations (panels d,e,f).

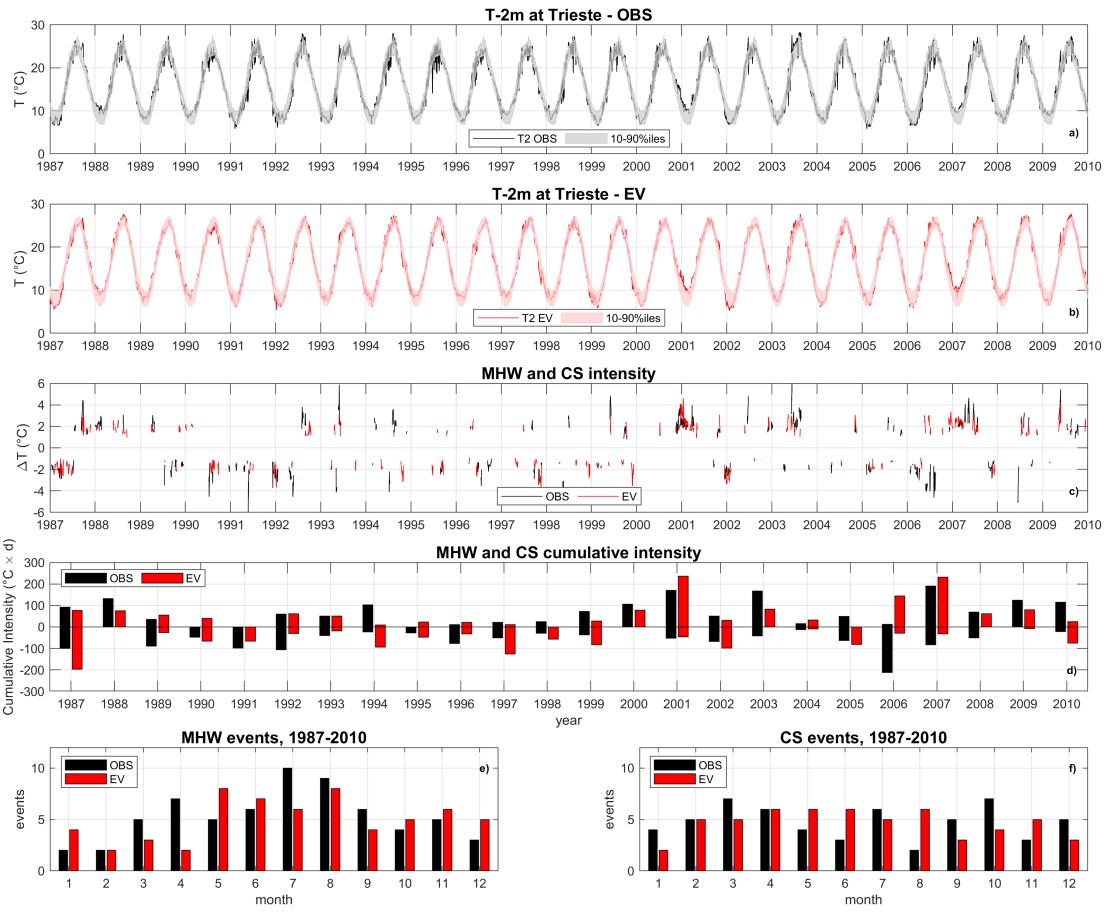

**Figure 13.** Comparison of modelled (EV) and observed thermal extreme events at Trieste station. Panels (a-b) represent respectively the observed and modelled time series alongside with the identification of the 10th and 90th daily percentiles for the period (here computed as a moving average within a 15-day sliding window); (c) highlights the events found in either series and their intensity; (d) compares the yearly cumulative intensity of the extreme events, and (e-f) compares for each month the observed and modelled number of MHWs and CSs respectively).

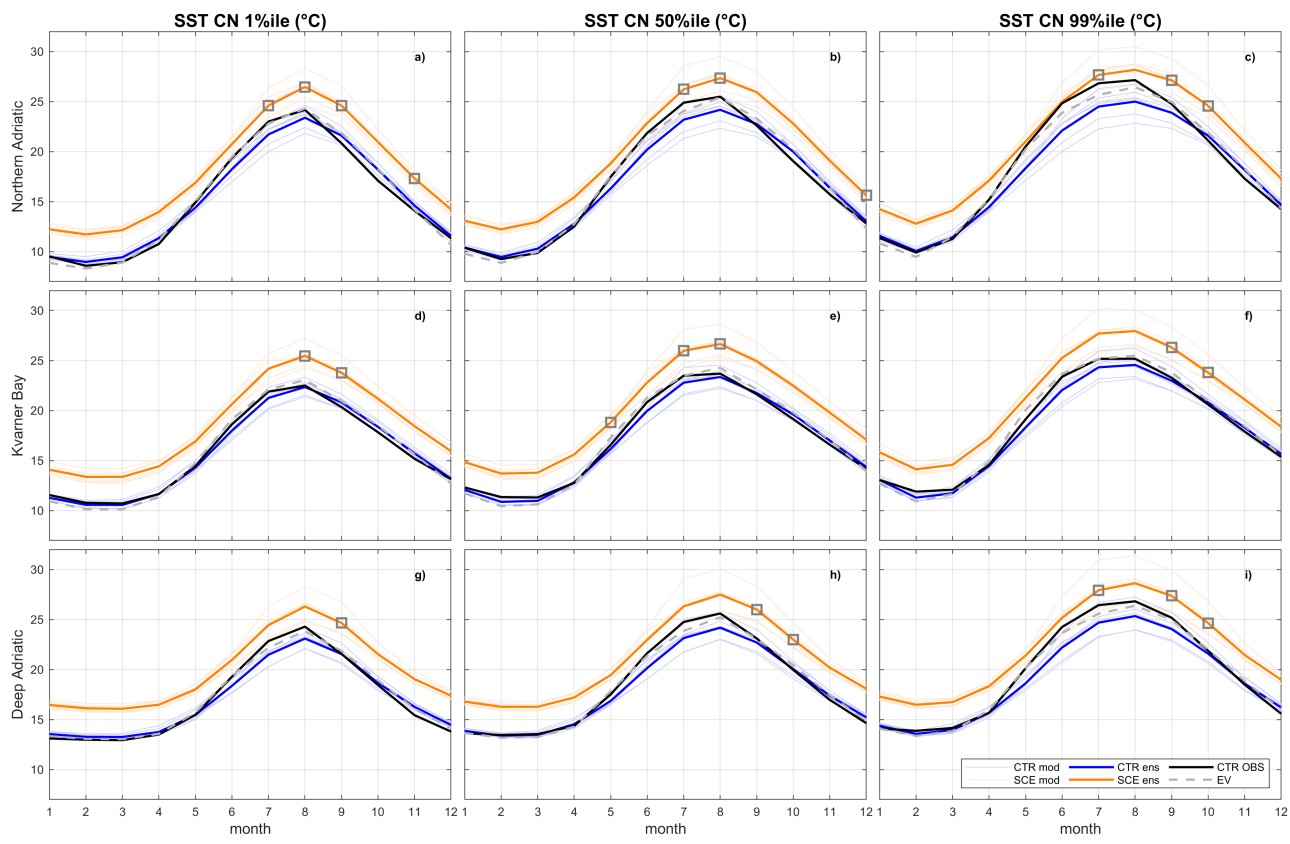

**Figure 14.** Modelled SST climate normals (namely, the average over the reference period) of different monthly statistics in different sub-basins under CTR (blue lines), and SCE (orange lines), where thin and thick lines represent respectively ensemble members mean, compared against observations in the historical period (CTR OBS, black thick line) and the evaluation run (EV, dashed gray line). Dark gray squares mark statistically non-significant variations in the ensemble distributions. Time segments: CTR/OBS 1987-2016; EV 1987-2010; SCE 2070-2099.

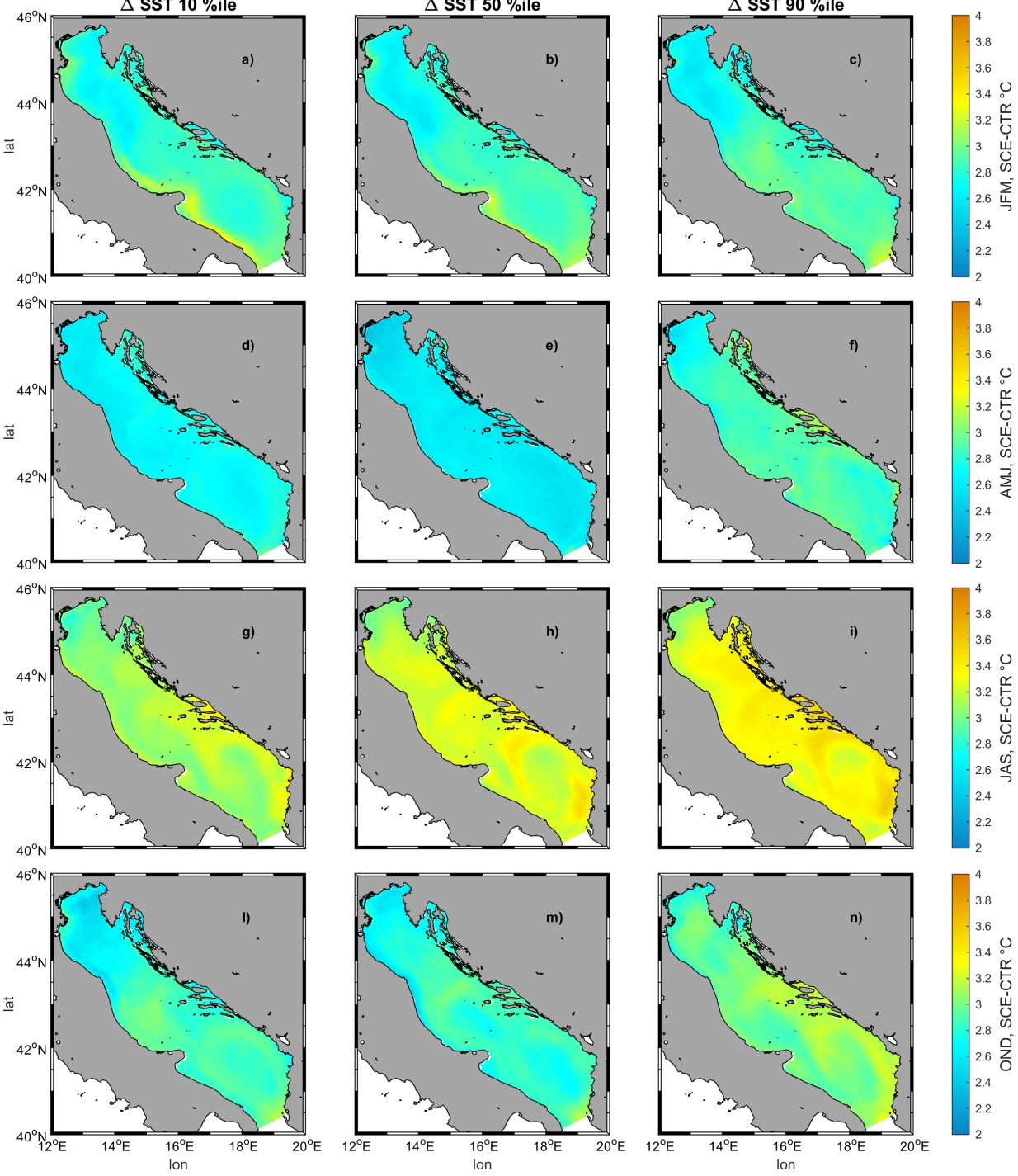

**Figure 15.** Seasonal variations (SCE-CTR) of ensemble mean sea surface temperature percentile climate normals. All displayed values are statistically significant.

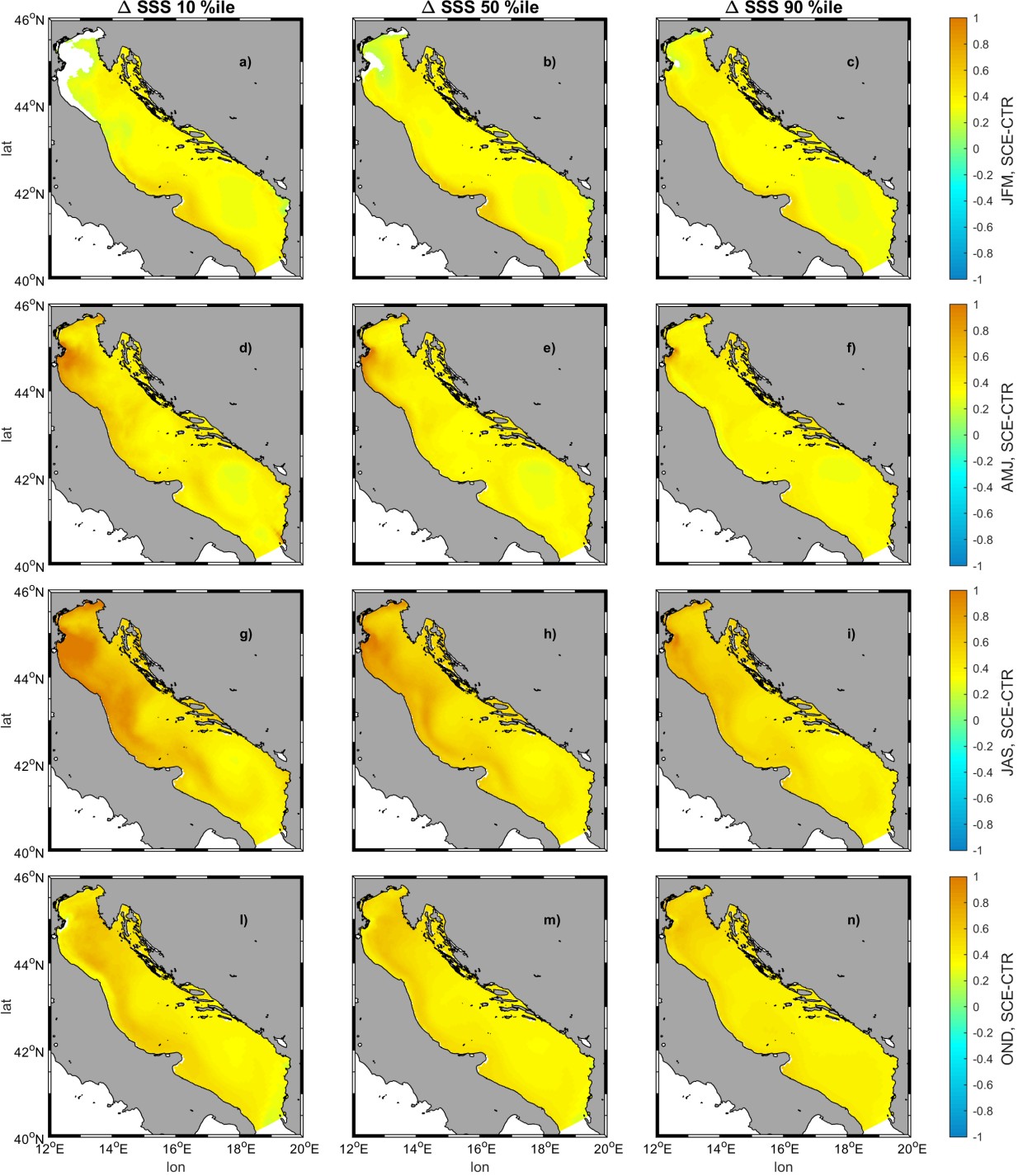

**Figure 16.** Seasonal variations (SCE-CTR) of ensemble mean sea surface salinity percentile climate normals. Blank regions indicate statistically non-significant differences.

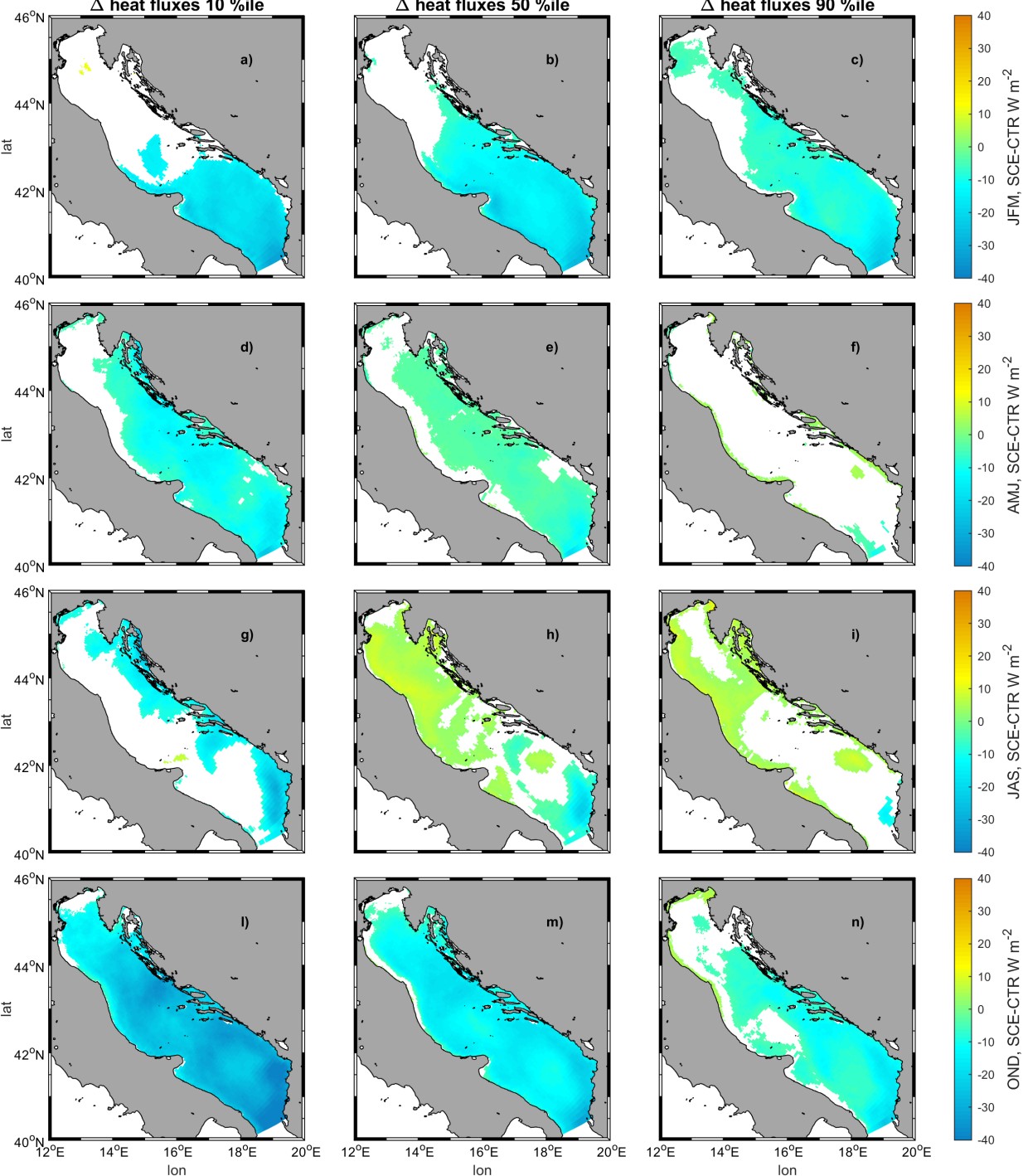

**Figure 17.** Seasonal variations (SCE-CTR) of ensemble mean sea net surface heat fluxes (positive values representing a heating influx) percentile climate normals. Blank regions indicate statistically non-significant differences.

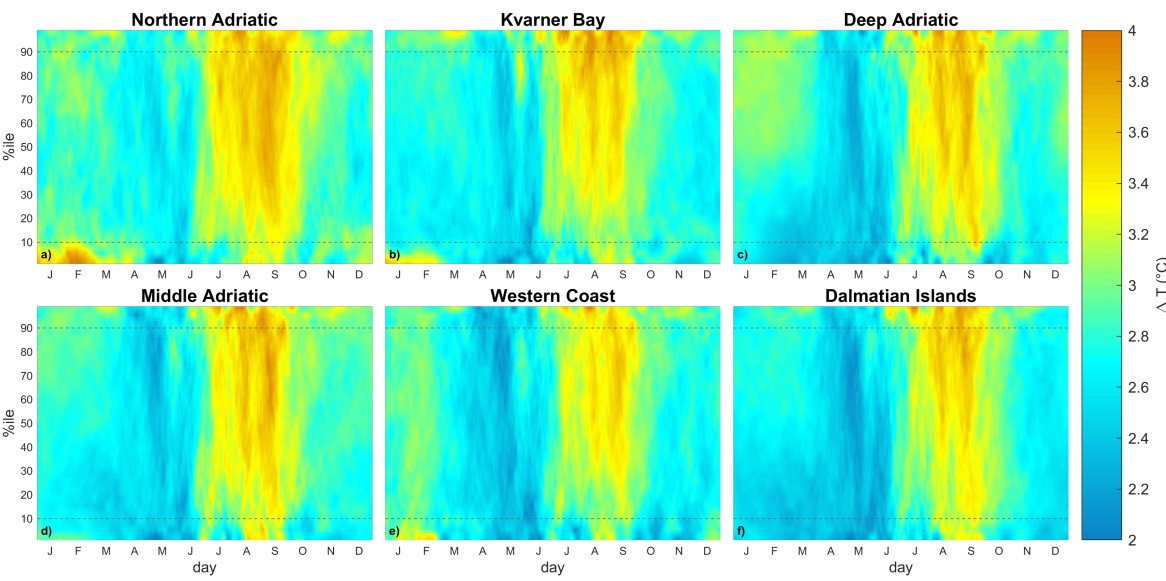

**Figure 18.** Comparison (SCE-CTR) of daily SST statistics in different subdomains. All displayed values are statistically significant.

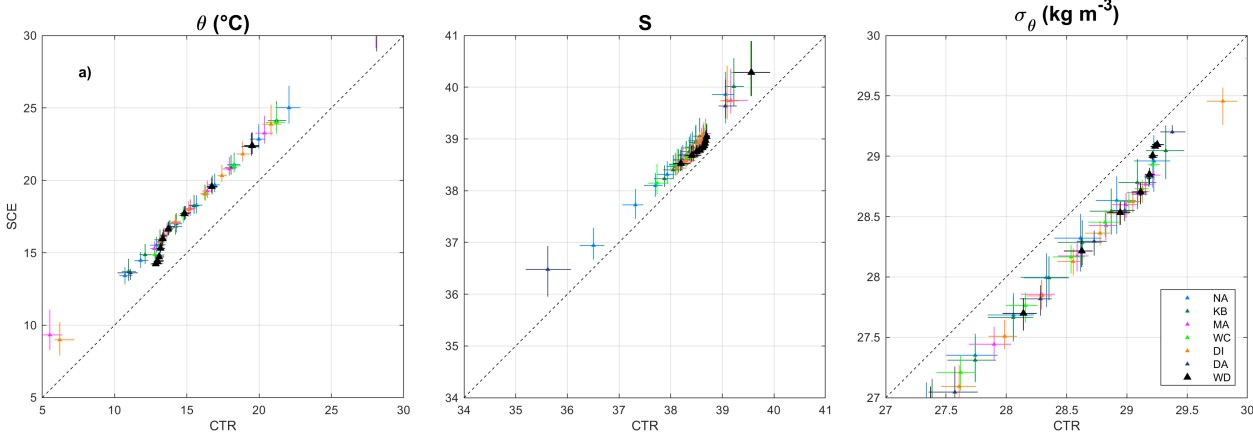

**Figure 19.** QQ plots representing CTR vs SCE ensemble statistics every 10 quantiles in different subdomains for potential temperature ($\theta$), salinity ($S$), and potential density anomaly ($\sigma_\theta$). Markers represent the ensemble mean for each considered quantile, and vertical and error bars represent the ensemble spread. The subdomains, identified in Figure 1b, are Northern Adriatic (NA), Kvarner Bay (KB), Middle Adriatic (MA), Western Coast (WC), Dalmatian Islands (DI), and Deep Adriatic (DA); WB represents the Whole Basin.

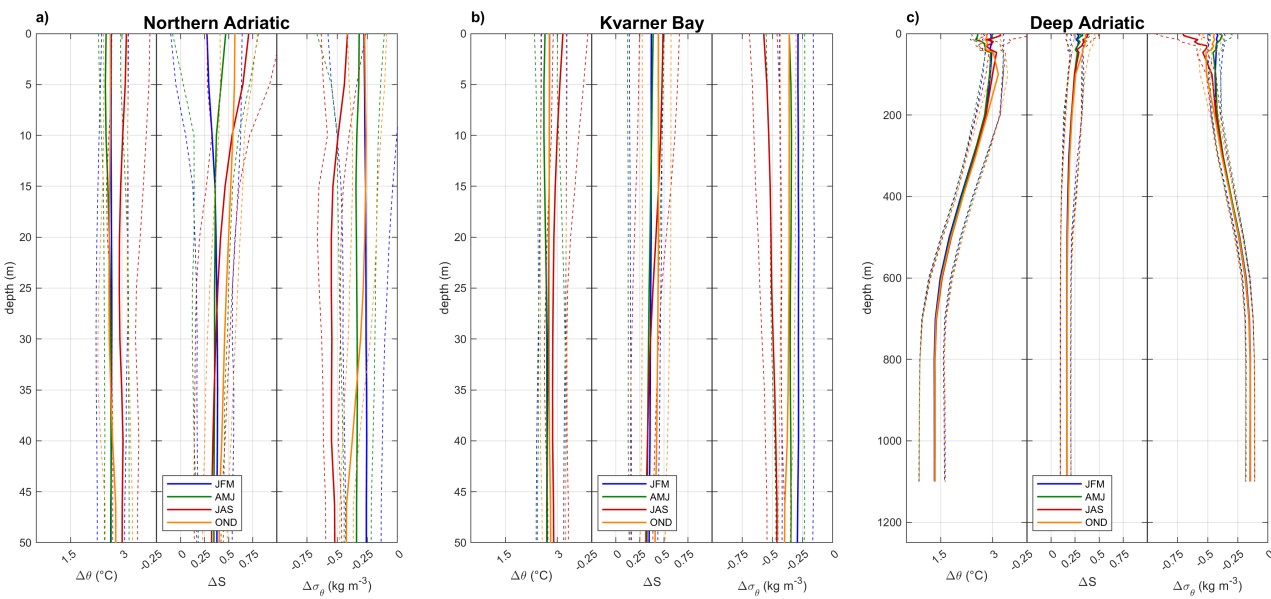

**Figure 20.** Variations (SCE-CTR) between median seasonal (again, with seasons defined as in Pranić et al. 2021 to enable the comparison with Figure 12) potential temperature ($\theta$), salinity ($S$), and potential density anomaly ($\sigma_\theta$) profiles in the Northern Adriatic (a), Kvarner Bay (b), and Deep Adriatic (c). Dashed lines bracket the ensemble spread.

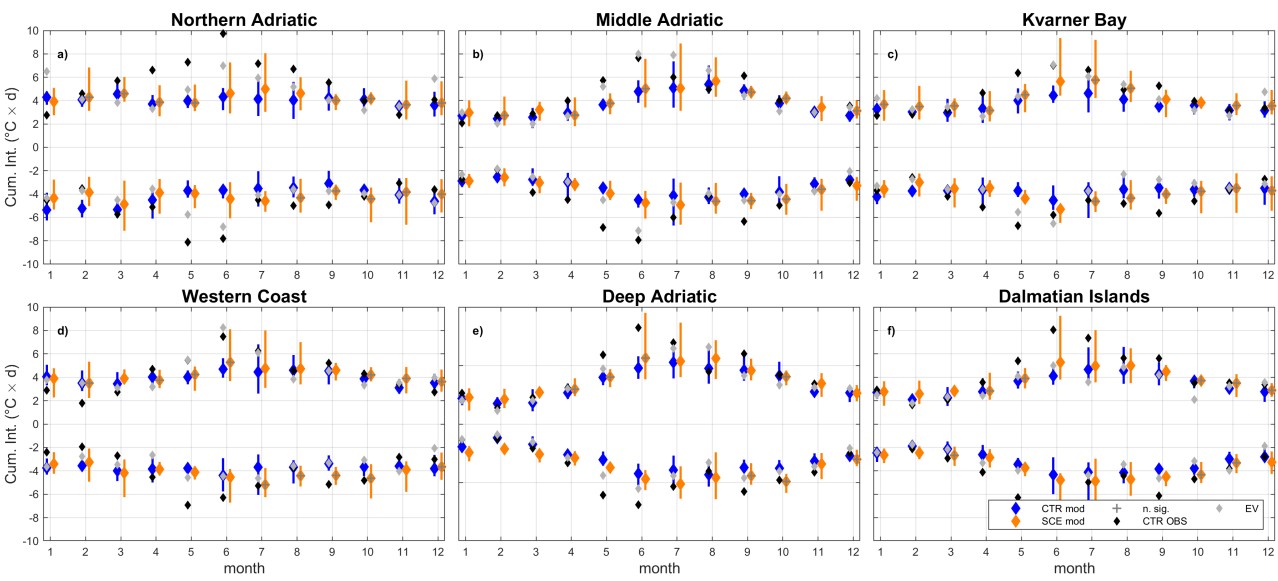

**Figure 21.** Monthly mean cumulative intensity of MHWs (>0) and CSs (<0) in CTR (blue markers) and SCE (orange markers), where vertical lines represent the ensemble spread, compared against the historical period (black markers) and the evaluation run (gray markers).Dark gray markers represent statistically non-significant variations in the ensemble distributions.