# Peer review of "AdriE: a high-resolution ocean model ensemble for the Adriatic Sea under severe climate change conditions"

_EGUsphere, 2024_

## Author Comment (AC1)

**RC1, 'Comment on egusphere-2024-1468', Anonymous Referee #1, 18 Jul 2024. Citation: https://doi.org/10.5194/egusphere-2024-1468-RC1**

The manuscript mostly concentrated on the Evaluation Run and its statistical properties by comparing the ocean model output with the available observations (SST, SLP, T). However, the manuscript missing a detailed analysis of the climate runs which is the main promise of the manuscript. The statistical analysis made for the Evaluation Run (Atmospheric forcings, Sea level variability and circulation patterns, Thermohaline properties, Extreme thermal events) should be repeated for all the climate runs.

*In this work the model validation was given a large attention as the idea was not only to assess the validity and limitations of the finding on the climate part, but also to provide some guidance on the use of the ensemble dataset associated with the present paper. Nonetheless, we concur on the opportunity of having a broader view on the climate results and we will expand this section for some relevant processes/quantities.*

*In the framework of this expansion it is important to notice (and we make it clearer in the revised version, as in the first one it was probably not sufficiently explicit) that the methodology for evaluation runs does not fully apply to the historical runs. While the former are driven by a reanalysis, which means that the RCM forcing the ocean model receives as boundary conditions reanalysis fields constrained to the observed atmospheric variability, the latter are driven by GCM fields, which result from a free run incorporating the observed variability "only" in terms of radiative fluxes (consistently with the constraints that can/cannot be controlled in a climate perspective) and generating the internal atmospheric variability as a free evolution under this condition. As a consequence, while atmospheric variability in reanalysis-driven runs is synchronised with the observed variability (and therefore it is licit to perform a direct one-to-one comparison of model and observations – that is, for instance, modelled fields for one day should match observed data for the same day), in GCM-driven runs this is only valid in statistical terms, and the modelled variability is expected to only match the statistical properties of atmospheric processes corresponding to a given radiative forcing regime. This is the reason why some assessment of the performance of the climate runs was actually included in the first version (Figures 13 and 16) only in aggregated terms.*

*On this ground, and with an eye on keeping the manuscript within a reasonable length, in the revised version we will expand along the following lines:*

- *Comparison of wind statistics in the climate run in CTR conditions against observations (*Figure 1*)*
- *Seasonal ensemble variations (percentiles) between SCE and CTR conditions in sea surface temperature (*Figure 2*) and net surface heat fluxes (*Figure 3*) .*
- *Differences in the statistical distribution of sea surface temperature and net heat fluxes for the different subdomains (*Figure 4*, *Figure 5*), particularly in the perspective of supporting the interpretation of the results in terms of thermal extremes (see also comments by Rev. 2 and our response).*

As it is known, the open boundary conditions (OBCs) for the ocean models are critical, especially in the Adriatic Sea, the small differences in the salinity and temperature specified at the OBCs significantly affect the dense water formation and physical properties. In the manuscript, how

the OBC data were generated to force the ROMS model is not clear, need to be specified the methods and justified that could be used safely in a climate model.

*We thank the reviewer for pointing this out. In the revised version we will expand the "model setup" Section with a more detailed explanation of how we imposed the boundary conditions and their modulation in the future scenario, and some comments on their suitability for the purposes of this study. Flanking the description of the methodology adopted we include a new figure (*Figure 6 *in this document) showing potential temperature and salinity distributions along the boundary cross section together with the velocity contours, also considering the EV\* run. In order to check that the prescribed climatological variations are realistic, we also include a comparison of the average trends, finding that the multidecadal tendency for potential temperature in the climate run in the historical period is well bracketed between the evaluation values (namely, the CMEMS reanalyses used as boundary conditions), while salinity trends (less clear in the reanalysis) appear slightly underestimated in the climate runs. The thermohaline properties appear consistent with typical values from the literature, particularly in terms of Modified Levantine Intermediate Water (MLIW, see for instance Bonaldo et al., 2016, and references therein). The known cyclonic flow across the boundary cross section is weaker in EV than in EV\*, but is recreated internally as geostrophic circulation (*Figure 8*), restoring the typical climatological values for MLIW around 0.10 m s$^{-1}$ (Orlić et al., 1992; Artegiani et al., 1997). Furthermore, the subdomain-based analysis of the evaluation run in the Results section has been expanded focusing on the surroundings of the model domain boundary (red polygon in Figure 7), and the Taylor diagram (Figure 9) now includes an assessment of the model in that region, showing a good agreement with measurements, with skill metrics comparable with the AdriSC reference (Pranić, et al., 2021).*

**References**

Artegiani, A., Bregant, D., Paschini, E., Pinardi, N., Raicich, F., Russo, A., 1997a. The Adriatic Sea general circulation. Part I: air–sea interactions and water mass structure. Journal of Physical Oceanography. 27, 1492–1514.

Bonaldo, D., Benetazzo, A., Bergamasco, A., Campiani, E., Foglini, F., Sclavo, M., Trincardi, F., Carniel, S. (2016). Interactions among Adriatic continental margin morphology, deep circulation and bedform patterns. *Marine Geology*, *375*. https://doi.org/10.1016/j.margeo.2015.09.012

Orlić, M., Gacić, M., La Violette, P.E., 1992. The currents and circulation of the Adriatic Sea. Oceanologica Acta 15 (2), 109–124.

Pranić, P., Denamiel, C., & Vilibić, I. (2021). Performance of the Adriatic Sea and Coast (AdriSC) climate component - A COAWST V3.3-based one-way coupled atmosphere-ocean modelling suite: Ocean results. *Geoscientific Model Development*, *14*(10), 5927–5955. https://doi.org/10.5194/gmd-14-5927-2021

[Figure]

[Figure]

*Figure 1: Revised version of Figure 4. Comparison of SMHI-RCA4 fields used: a) in the EV run, and b) in the climate ensemble (coloured bars) against CCMP directional wind statistics and in situ observations at AA.*

[Figure]

Figure 2: Seasonal variations (SCE-CTR) of the 10, 50, 90 percentile SST

[Figure]

Figure 3: Seasonal variations (SCE-CTR) of the 10, 50, 90 percentile net surface heat fluxes

[Figure]

*Figure 4: Variations (SCE-CTR) in the daily statistics for SST*

[Figure]

*Figure 5: Variations (SCE-CTR) in the daily statistics for SST*

[Figure]

Figure 6: Time-averaged boundary conditions (panels a, b, d, e) and trends (panels c, f). Thick lines and dotted lines represent 0.01 m/s velocity contours in the outflow and inflow direction respectively.

[Figure]

Figure 7: revised version of Figure 1

[Figure]

*Figure 8: Revised version of Figure 6, adding circulation patterns in the 150-500 m depth range (panel c)*

[Figure]

*Figure 9: Revised version of Figure 10, including OS (Otranto Strait) as an additional subdomain alongside those defined in Pranić et al. 2021.*

---

## Author Comment (AC2)

**RC2: 'Comment on egusphere-2024-1468', Anonymous Referee #2. Citation: https://doi.org/10.5194/egusphere-2024-1468-RC1**

Review of the paper

The manuscript deals with the present and the end-of-century, kilometre-scale ensemble modelling approach for the description of ocean processes in the Adriatic Sea using and ensemble of climate runs in a severe RCP8.5 scenario forced by the SMHI-RCA4 Regional Climate Model driven by CMIP5 General Climate Models as well as evaluation runs for the 1987-2010 period.

The text is well written and results are presented clearly.

*We thank the reviewer for this positive comment*

The authors show that the main behaviour of the model used is ''satisfactory''. However scenario simulations show results that necessitate a deeper investigation of the role of the model set up and of the forcing. The choice of the lateral boundary conditions is also of a crucial importance.

A deeper investigation of the relatively high theta and S changes in the deep Adriatic is recommended. This is also the case of the weak change of MHWs shown.

*We do appreciate the reviewer's suggestion. In fact, our positiveness lies mostly in the fact that the model performance against observations is comparable with the one exhibited by a state-of-the-art hindcast (AdriSC, referenced in the manuscript), which is a very good result in a climate model. The purpose of this manuscript and of the associated dataset is to pave the way to a number of studies on the many processes that take place in the Adriatic Sea. In principle the role of different factors (and in particular of atmospheric forcings and boundary conditions) depends on the process to be investigated (for instance, a study on river plume spreading in future conditions would not depend on the same processes and metrics as a study on marine heat waves or on dense water formation), and an extensive discussion on each of these aspects is beyond the scope of this manuscript and probably unfeasible for a single paper. In this direction, the scope of the validation presented in the manuscript lies mostly in presenting the potential of the model and its dataset, the possible limitations, and in discussing the results presented in the climate scenarios. In the revised version we will be clearer in framing these objectives and recalling the need of a specific analysis and discussion of forcings and boundary conditions in the future applications.*

*Nonetheless, we agree with the reviewer that a somewhat deeper (though with an eye on keeping the manuscript to a reasonable size) general discussion on boundary conditions, atmospheric forcings, their role on the results and the implications for other applications would be beneficial for the paper and helpful in the use of the dataset, and we will follow the reviewer's suggestion. In the revised version we will present in the "Model Setup" section a more detailed explanation of how the boundary conditions were introduced and a check on the trend of the values prescribed in the climate simulations (reflecting the variability of the CMCC-CM profiles) against the ones prescribed in the EV and EV\* runs (reflecting the two versions of the CMEMS reanalysis), while a quantitative assessment of their quality is introduced in the Results in an updated version of the Taylor diagrams (see* Figure 1 *and* Figure 2 *in the present document). We will point out that the thermohaline properties appear consistent with typical values from the literature, particularly in terms of Modified Levantine Intermediate Water (MLIW, see for instance Bonaldo et al., 2016, and*

*references therein), and the known cyclonic flow across the boundary cross section is weaker in EV than in EV\* but in any case is recreated internally as geostrophic circulation (see also response to Rev. 1), restoring the typical climatological values for MLIW around 0.10 m s$^{-1}$ (Orlić et al., 1992; Artegiani et al., 1997). Furthermore, the Taylor diagram (*Figure 2*) shows a good agreement with measured values, again with skill metrics comparable with the AdriSC reference (Pranić, et al., 2021).*

*In addition, also in the direction of expanding the "climate" part of the manuscript as suggested by Rev. 1, we introduced some additional results from the climate runs.*

*An assessment of the wind regimes in the historical part of the climate runs (*Figure 3*) shows a good match with observations at sea in the Northern Adriatic also for the GCM-driven simulations, suggesting that overall realistic wind regimes are used also as a forcing for the climate runs.*

*The variations of ensemble seasonal percentiles between SCE and CTR conditions for sea surface temperature (*Figure 4*) and net surface heat fluxes (*Figure 5*), as well as the differences in the daily climatological CDFs for the same quantities (*Figure 6, Figure 7*) in the different subdomains are introduced mainly to enhance the analysis of the climate variability and the discussion of the results on thermal extremes. Nonetheless, complemented with the trends on the boundary conditions (*Figure 1*, panels c and f), they will be used to draw some considerations on the effect of surface fluxes and boundary conditions in the basin properties, although again recalling that fully disentangling the role of each factor in general is beyond the scope of this work.*

*The discussion will move along these lines, addressing the reviewers questions raised above. In particular, we anticipate here that the values of theta and S changes, consistent with the trends imposed at the boundary and modulated by the internal dynamics throughout the basin, will be discussed also referring to other climate projections available in the literature for this region. Instead, the weak change in MHWs and CSs seems consistent with the limited change in the shape of the statistical distribution of SST around the extremes (*Figure 6*): in particular, considering that here we take as a threshold the 90$^{th}$ and 10$^{th}$ percentiles of each period in which we compute the extremes, we would expect that major differences in MHWs and CSs would be associated with larger differences in extreme values than around those thresholds, which does not seem to be the case. It should also be pointed out (and we will do it in the revised version) that, concerning the atmospheric forcing, there is increasing evidence of a generalised tendency of GCMs and RCMs to underestimate temperature extremes, mostly due to the neglection of aerosol changes (see for instance Schumacher et al., 2024). In any case, we will also comment more extensively on the implications of defining future MHWs and CSs based on the future statistics instead of the present thresholds.*

I recommend major revisions.

 Specific comments:

-15. '' with particularly encouraging results''

Could authors explain to what extent results are encouraging?

*The overall good capability of the model to reproduce the main features of observed Marine Heat Waves (MHWs) and Cold Spells (CSs), such as timing, intensity, and interannual variability suggests that our dataset could effectively be used for studies involving thermal extremes (e.g. linked to ecological processes, etc.). We rephrase this to make it clearer in the revised version.*

-45. '' (?Denamiel et al., 2021a) ''

Please correct if needed.

*Thanks, fixed*

-50. '' ... ranging from the very evolution of the global...''

Could authors verify this sentence?

*Thanks, modified into "...from the evolution of the global climate to how this signal propagates through different scales, and how the adopted numerical description impacts the final results".*

-100. ''Potential temperature ($\theta$), salinity (S), momentum...''

Could authors describe how momentum is used in the boundary condition set up?

*In the revised version we will provide more details on the boundary conditions. In particular, we will point out that we imposed Chapman conditions (Chapman, 1985) for free surface, Flather conditions (Flather, 1976) for 2D momentum components, and nudged radiative conditions for 3D momentum components and tracers (potential temperature and salinity).*

-105. ''...were modulated accordingly with the anomalies computed from Med-CORDEXderived CMCC-CMprofiles (Scoccimarro et al., 2011) in the norhtheasternmost grid cell of the Ionian Sea.''

Could authors better describe the approach followed?

*Thanks, in the revised version we reshape the description along these lines: "For the EV run, daily reanalysis values were directly interpolated on the model grid points throughout the cross section. For the climate runs, climatological monthly values were first computed from the reanalysis fields with reference to the 1987-2017 period. These values were then perturbed with the anomalies computed, with reference to the same period, from Med-CORDEX derived CMCC-CM profiles (Scoccimarro et al., 2011) in the northeasternmost grid cell of the Ionian Sea."*

-185. ''Furthermore, although being the only available option the evaluation of SMHI-RCA4, ERA-INTERIM known to be far from the "perfect boundary conditions" hypothesis, particularly in terms of rainfall-related quantities (Bao and Zhang, 2013).''

This sentence is rather unclear, could authors rephrase it.

*We thank the reviewer for pointing this out, actually we were missing a verb! We apologise for that. We add the verb and slightly modify the sentence into "Furthermore, although being the only available option the evaluation of SMHI-RCA4, ERA-INTERIM does not presently represent the state of the art for atmospherical modelling, and is known to be far from the "perfect boundary conditions" hypothesis, particularly in terms of rainfall-related quantities (Bao and Zhang, 2013)".*

-210. Whereas the comparisons shown in Fig.2, Fig.3 and 4 show that the model behaviour is rather satisfactory as stated by the authors: ''thus performing significantly better than most of the RCMs available for this geographical area'', it would be interesting to illustrate this by one or two concrete examples.

*We thank the Reviewer for helping us noticing that our reference to the RCA4 regional climate model (RCM) skill were probably slightly overenthusiastic. We have refined this phrase removing "thus performing significantly better than most of the RCMs available for this geographical area" from the original sentence. In fact, strictly speaking this actually represents an overstatement since a specific assessment based on a multi-RCM comparison over this particular region and variables lies outside the scope of the study. Nevertheless RCA4 shows overall representative skills for essential climate variables as preliminarily assessed in the context of a previously published article and involving similar geographical domain (Bonaldo et al., 2023) as well as in review articles, including the large CORDEX ensemble (Coppola et al., 2021; Diez-Sierra et al., 2022; Vautard et al., 2021), and specifically over the Adriatic region where Belušić Vozila et al., (2019) consider wind climate variable specifically. The references have been added to the revised version of the manuscript.*

*Another element driving us towards using this model is that at the time in which climate simulations were gathered, RCA4 was the RCM with the largest number of simulations (corresponding to different driving GCMs) with a sub-daily time frequency for quite a large number of variables, required for driving the ROMS model. As outlined in the methodology section, this approach is appealing within the research framework, as it aims to limit uncertainty sources by focusing on a single RCM setup rather than exploring multiple RCMs driven by different GCMs.*

-230. "This northbound improvement of the model skills suggests that internal dynamics partially compensate for the missing variability component in the boundary conditions." ; "...the fairly good performance on the Northern Adriatic coast permits a more straightforward use in this region, also in terms of boundary conditions for local applications".

-What is the sampling interval used in Fig.5?

-Are tidal oscillations included in Fig .5?

-The tidal amplitude is known high in the northern Adriatic; could authors discuss the impact of the tidal amplitude of the model performance shown in Fig.5.

-Again authors should further discuss the lateral boundary conditions.

*Although tides were included in the EV and EV\* runs by considering 15 tidal components from the TPXO dataset (Egbert and Erofeeva, 2002), Figure 5 (in the manuscript) refers to daily-averaged data, and therefore semi-diurnal and diurnal components are lost, and the underestimate in sea surface level variability shown in Figure 5 does not include tidal variability. In a deeper analysis*

*not included in this manuscript for the sake of synthesis we found that tidal variability resulting from TPXO is generally underestimated. This may contribute to some mismatch in circulation and tracer transport patterns over the short term, but since the result of the validation is considered in aggregated terms, we don't see obvious reasons to expect systematic errors introduced by this factor, whilst it most likely contributes to add some noise around the average skills.*

*This is better explained in the revised version.*

-270. ''... , suggesting that SST does not show any macroscopic sign of a spurious drift related to the model implementation ".

 Could authors rephrase this sentence?

*Thanks, in this form this sentence aimed at addressing the (undeclared) possibility that a bias in the flux parameterizations at the air-sea interface or numerical issues could, at the multi-decadal time scale, result in an unrealistic temperature drift. In fact, for this sentence to be clear it would call for a better discussion of this possibility, but then it would probably burden the discussion without adding an important contribution (in the end, the drift is not there!). We thus decided to remove this sentence.*

-280. '' In turn, while intermediate to high S values are mostly well reproduced, low to mid salinity tends to be overestimated, particularly in the Kvarner Bay and in the Dalmatian Islands.''

Please better discuss and explain the overestimation of the lowest S values.

*Thanks, two main aspects can have a role in this result. First, the estimate of submarine freshwater inputs in the karstic northeastern and eastern Adriatic coast is a recognisedly challenging task in the area, and can lead to significant uncertainties. Secondly, the model resolution does not permit a complete description of the complex geomorphology of that coast, and therefore of its small-scale circulation patterns. This is better explained in the revised version.*

-285. '' This suggests that the climate ensemble, whose implementation began before the release of the latest version of MFS (Escudier et al., 2020), should not be considered prone to major elements of obsolescence associated with the use of a previous dataset (Simoncelli et al., 2019)".

Could authors explain how this can be deduced from Fig.10.

*Here the idea is that, if the use of the latest version of MFS does not significantly improve the overall skill metrics, it seems reasonable to expect that the use of a previous dataset (the only one available at the time of the ensemble implementation) to compute the climatologies at the boundary should not lead to major shortcomings in the climate runs. We better frame this concept in the revised version.*

-305. '' In the deep Adriatic, an apparent tendency to underestimate average values of θ and S throughout the year is actually the result of some shortcomings in the description of thermohaline properties in the upper layers.''

The sentence needs further explanation of the mentioned shortcomings.

*Thanks for this suggestion, in fact the sentence should be adjusted. More precisely, the first part refers to panel c, but we do not have enough elements to actually attribute this result to some processes, or shortcomings taking place in the upper layers. Instead, while there is certainly an overestimate of heating and mixing in the upper layers, the observed underestimate at deeper layers could be inherited from the dataset used for initialization and boundary conditions (the results presented by Pranic et al. 2021, which were based on the same datasets, showed very similar values). We will make this clearer in the revised version, though pointing out that a conclusive interpretation of this mismatch requires a dedicated effort.*

-370. Please correct : ''hle200 m''

*Thanks, adjusted.*

-375. ''Below the upper layer, θ increase varies from + 2.8°C for h=200 m to +1.3°C for h≥800 m, S increase varies from +0.21 to +0.17, and σθ varies between-0.44 and-0.15 kg m−3.''

Authors should discuss the relatively high values of salinity and theta changes in the deep Adriatic (shown in Fig.15c) and present comparison with results from previous work. Also, why two among the vertical profiles of the theta change are truncated at depths less than ~630 m,  (Fig. 15c left).

*The truncation of the bottom values of the ensemble spread resulted from the graphical setting for the x-axis limits, we adjust it in the revised version. The values of the temperature and salinity variations in the Deep Adriatic are consistent with the trends prescribed at the boundary (Figure 1 of this document). We will comment along this line (Figure 1 will actually be included in the revised version) also in the light of the available literature.*

-385. ''Under this approach, modelled differences between SCE and CTR conditions (expressed as monthly mean cumulative intensity of the events) appear generally minor and in any case only occasionally statistically significant.''

Here also, authors should mention results from previous work, if available.

*Thanks, we will check for latest literature in this direction, and discuss if possible.*

**References**

Artegiani, A., Bregant, D., Paschini, E., Pinardi, N., Raicich, F., Russo, A., 1997a. The Adriatic Sea general circulation. Part I: air–sea interactions and water mass structure. Journal of Physical Oceanography. 27, 1492–1514.

Belušić Vozila, A., Güttler, I., Ahrens, B., Obermann-Hellhund, A., & Telišman Prtenjak, M. (2019). Wind Over the Adriatic Region in CORDEX Climate Change Scenarios. *Journal of Geophysical Research: Atmospheres*, *124*(1), 110–130. https://doi.org/10.1029/2018JD028552

Bonaldo, D., Bellafiore, D., Ferrarin, C., Ferretti, R., Ricchi, A., Sangelantoni, L., & Vitelletti, M. L. (2023). The summer 2022 drought: a taste of future climate for the Po valley (Italy)? *Regional Environmental Change*, *23*(1). https://doi.org/10.1007/s10113-022-02004-z

Coppola, E., Nogherotto, R., Ciarlo', J. M., Giorgi, F., van Meijgaard, E., Kadygrov, N., et al. (2021). Assessment of the European Climate Projections as Simulated by the Large EURO-CORDEX Regional and Global Climate Model Ensemble. *Journal of Geophysical Research: Atmospheres*, *126*(4), 1–20. https://doi.org/10.1029/2019JD032356

Diez-Sierr, J., Iturbide, M., Gutiérrez, J. M., Fernández, J., Milovac, J., Cofiño, A. S., et al. (2022). The Worldwide C3S CORDEX Grand Ensemble A Major Contribution to Assess Regional Climate Change in the IPCC AR6 Atlas. *Bulletin of the American Meteorological Society*, *103*(12), E2804–E2826. https://doi.org/10.1175/BAMS-D-22-0111.1

Chapman, D.C., 1985. Numerical treatment of cross-shelf open boundaries in a barotropic coastal ocean model. Journal of Physical Oceanography 15, 1060‑1075.

Bonaldo, D., Benetazzo, A., Bergamasco, A., Campiani, E., Foglini, F., Sclavo, M., Trincardi, F., Carniel, S. (2016). Interactions among Adriatic continental margin morphology, deep circulation and bedform patterns. *Marine Geology*, *375*. https://doi.org/10.1016/j.margeo.2015.09.012

Flather, R.A., 1976. A tidal model of the northwest European continental shelf. Memoires de la Society Royal des Sciences de Liege 6, 141‑164.

Orlić, M., Gacić, M., La Violette, P.E., 1992. The currents and circulation of the Adriatic Sea. Oceanologica Acta 15 (2), 109–124.

Pranić, P., Denamiel, C., & Vilibić, I. (2021). Performance of the Adriatic Sea and Coast (AdriSC) climate component - A COAWST V3.3-based one-way coupled atmosphere-ocean modelling suite: Ocean results. *Geoscientific Model Development*, *14*(10), 5927–5955. https://doi.org/10.5194/gmd-14-5927-2021

Schumacher, D. L., Singh, J., Hauser, M., Fischer, E. M., Wild, M., & Seneviratne, S. I. (2024). Exacerbated summer European warming not captured by climate models neglecting long-term aerosol changes. Communications Earth and Environment, 5(1). https://doi.org/10.1038/s43247-024-01332-8

Scoccimarro, E., Gualdi, S., Bellucci, A., Sanna, A., Fogli, P. G., Manzini, E., Vichi, M., Oddo, P., and Navarra, A.: Effects of Tropical Cyclones on Ocean Heat Transport in a High-Resolution Coupled General Circulation Model, Journal of Climate, 24, 4368 – 4384, https://doi.org/10.1175/2011JCLI4104.1, 2011.

Vautard, R., Kadygrov, N., Iles, C., Boberg, F., Buonomo, E., Bülow, K., et al. (2021). Evaluation of the Large EURO-CORDEX Regional Climate Model Ensemble. *Journal of Geophysical Research: Atmospheres*, *126*(17). https://doi.org/10.1029/2019JD032344

[Figure]

Figure 1: Time-averaged boundary conditions (panels a, b, d, e) and trends (panels c, f). Thick lines and dotted lines represent 0.01 m/s velocity contours in the outflow and inflow direction respectively.

[Figure]

*Figure 2: Revised version of Figure 10, including OS (Otranto Strait) as an additional subdomain alongside those defined in Pranić et al. 2021.*

[Figure]

[Figure]

*Figure 3: Revised version of Figure 4. Comparison of SMHI-RCA4 fields used: a) in the EV run, and b) in the climate ensemble (coloured bars) against CCMP directional wind statistics and in situ observations at AA.*

[Figure]

*Figure 4: Seasonal variations (SCE-CTR) of the 10, 50, 90 percentile SST*

[Figure]

Figure 5: Seasonal variations (SCE-CTR) of the 10, 50, 90 percentile net surface heat fluxes

[Figure]

Figure 6: Variations (SCE-CTR) in the daily statistics for SST

[Figure]

Figure 7: Variations (SCE-CTR) in the daily statistics for SST

---

## Author Comment (AC3)

**CC1** 'Comment on egusphere-2024-1468', Iana Strigunova, 10 Oct 2024

The manuscript introduces a kilometre-scale ensemble modelling approach to ocean processes in the end-of-century (RCP8.5) scenario in the Adriatic Sea. The modelling chain ensemble is achieved by utilising the ROMS modelling system forced by the SMHI-RCA4 Regional Climate Model, driven by five different climate models and two different realisations of the same model.

While the current version requires modifications, its scientific value (considering trade-offs between high resolution and decadal climate projections) and the region's importance as a climate change hotspot, for instance, are evident. Please find the suggestions for the manuscript's improvement below.

*Thank you very much for carefully reading our manuscript and taking the time to participate in the discussion. You suggestions are most appreciated and they will definitely improve the quality of the result.*

The 3.1.4 subsection ('Extreme thermal events') discusses the evaluation of thermal extremes identified from Trieste harbour station (Fig. 12). While the comparison is completely valid for one point, it does not provide an understanding of how these extremes are simulated for the subdomains of the Adriatic Sea, which are used to describe the difference in the statistics of the present and future MHWs and CSs (Fig. 16).

*Thanks for pointing this out. Taking your suggestion, we have repeated the calculation for each subdomain using the sea surface temperature observations from from the L4 Optimal Interpolation (L4OI) Mediterranean Advanced Very High Resolution Radiometer (AVHRR) SST Analysis dataset (Pisano et al., 2016). The results (Figure 1 to Figure 8) essentially confirm the findings from the pointwise observation in Trieste and, although for the sake of brevity we will not include this set of plots in the revised version, we will mention that this verification was carried out and the results shown for Trieste are representative of the performance in the whole domain.*

**Minor comments**

The study greatly explores the possibility of an ensemble modelling approach for the Adriatic Sea. Could these results apply to other parts of the Mediterranean Sea or even to other global regions? By addressing this question, the manuscript may be attractive for more readers.

Some parts of the abstract seem unnecessarily wordy, making them hard to read and potentially hampering readers' understanding of the study's real significance. For instance, while the abstract's first two sentences extensively discuss the study's importance, they do not clearly explain why it is essential to study the Adriatic basin or why this study could interest researchers not directly dealing with the Adriatic Sea. In this regard, I would suggest two changes: 1) please consider moving these sentences to the introduction and 2) perhaps rephrase it in a bit more concise manner so readers familiarise themselves with the study's significance quickly. The clarity of the abstract could be improved by starting from the sentence in line 9 ('This work presents...').

*Thank you for this suggestion, in the revised version we will sharpen the abstract and provide a wider view on the possible implications of this study beyond its regional setting. In this direction, we will highlight that, alongside with useful elements from the methodological/modelling point of view, this study presents a large dataset that can be used for the analysis of coastal and*

*continental margin processes of general interest, e.g. dense water dynamics and the role of the interplay of different factors such as basin preconditioning, buoyancy fluxes and circulation patterns in its formation and spreading.*

Lines 6-9: "…a description of the possible evolution of the physical oceanographic processes is the baseline for addressing the multi-disciplinary challenges set by climate change, … ". I would be more cautious stating in this way. It is one of the key processes, but others are no less important (biogeochemistry processes and human influence, for instance).

*Thanks, "baseline" will be changed into "one of the key requirements" in the revised version*

Line 33: "Due to the coexistence of manifold meteo-oceanographic processes…". It is not clear what it implies. Perhaps readers who are not familiar with specific regional features will not understand the importance.

*Thanks for pointing this out, in the revised version we will briefly outline the main processes that characterize the Adriatic Sea making it a "natural laboratory for marine science".*

Lines 38 and 153: I am unfamiliar with the 'EO analysis' and 'BiOS' abbreviations. Could you please clarify what does it mean?

*Here we use "EO" as an acronym for "Earth Observation", while BiOS refers to the Bimodal Ionian Oscillation, a decadal modulation of the Eastern Mediterranean circulation patterns affecting the pathways of water masses and in particular the exchanges between the Adriatic and Ionian seas (for a broad review on this concept I would suggest the recent work by Civitarese et al., 2023, and references therein). We clarify both these acronyms in the revised version.*

Lines 208-213: The paragraph logically flows from the previous ones, but there is no reference to Fig. 4, which is discussed in the previous paragraph. Would it be possible to change the order or add a part about Fig. 4 in the last paragraph for a more comprehensive summary and a following conclusion?

*Thanks for this suggestion, in the revised version we will add a reference to Fig. 4 in the conclusion of this paragraph.*

Lines 363-376. This paragraph has many numbers based on Fig. 15, which hinders understanding of what they actually mean. I would suggest modifying the paragraph in the following way: adding a summary table with numbers and modifying the text so that only the essential findings with no numbers are kept. That would allow to describe Fig. 15 and interpret the results in a clearer manner.

*Thanks for this suggestion. Actually, our feeling is that a summary table would not add much with respect to Fig. 15, and therefore could look somewhat redundant. In any case, in the revised version we will try and reshape this part aiming at improving its readability.*

Section 4 ('Conclusions') opens up with a summary of what the authors did. Starting from line 407, it is more of a discussion form until line 419. The main results are summarised in lines 425-439. Perhaps authors could allocate a separate section for lines 407-418 and keep all summary points together in the Conclusions to make it clearer for readers.

*Thanks for pointing this out. At present, the discussion part in this paragraph is mostly aimed at recalling the potential use and main limitations of this work and the associated dataset, and in our view it should fit with an overview on what was done, its strength points and caveats, and take-home messages. In any case, the narrative could and probably should be improved, and in the revised version we will take care of this.*

**Technical comments**

Lines 23-24 (440-442): Could you please specify why this type of information is repeated across the manuscript? Mentioning data availability on specified repositories and request to the corresponding author are finely placed in the "Data availability" section.

*This information is recalled as one of the purposes of this work is to open the way to a variety of future studies about climate change in the Adriatic Sea, and in this direction we strive to provide guidance and encourage the use of the AdriE dataset.*

Line 35: "… the presence of highly-exposed sites of outstanding natural and cultural value…". This part seems unnecessarily wordy. What do you mean by 'highly-exposed' sites? It is not completely clear.

*Here we refer to "exposure" in terms of the importance of the assets subject to coastal hazards. We will adjust this part to improve clarity and readability.*

It would be beneficial to extend Table 1 by adding information on horizontal resolution. Additionally, having each centre's name to the model's name (NCC, IPSL, MPI, etc.) appears redundant since they have already been used in the first column.

*Thanks for this suggestion. Actually the horizontal resolution is the same for all runs (0.11°), so a possible additional column would not really bring relevant information. Concerning the naming of the runs, true it is that there is some repetition between the "Run" and the "Driving GCM" column, but this does not hold for all runs (e.g. EV and EV\*), and in any case the "Driving GCM" reflects the CORDEX nomenclature allowing experiment repetition and/or expansion.*

Table 3: the "Name" column should be wider to enable each name to fit in one row.

*Thanks, this will be fixed in the revised version.*

Fig. 1 and 2 could also be reallocated to the rest of the figures or the other figures placed within the main text for consistency.

*Thanks for this suggestion. In the revised version we will provide a more consistent positioning of the figures, although we expect that the final adjustment will be performed in the copyediting phase if the manuscript is accepted for publication.*

The caption of Fig. 1: Maybe I missed it, but have you introduced the 'AS ' abbreviation?

*Thanks for pointing this out, the abbreviation can definitely be removed and in the revised version we just refer to "Adriatic Sea".*

Line 198: '...(>15 ms−1, see Mears et al. 2022) wind speed)...' it seems the extra ')' symbol can be removed.

*Thanks, fixed.*

Line 278: "*theta*" Please correct if needed.

*Thanks, this has been corrected.*

Line 315: "... of Marine Heat Waves (MHWs) and Cold Spells...' to "... of Marine Heat Waves (MHWs) and Cold Spells (CS) ...' Please correct if needed.

*Thanks, corrected into "Marine Heat Waves (MHWs) and Cold Spells (CSs)".*

Line 353: "σ*theta*" Please correct if needed.

*Thanks, corrected.*

**References**

Civitarese, G., Gačić, M., Batistić, M., Bensi, M., Cardin, V., Dulčić, J., Garić, R., & Menna, M. (2023). The BiOS mechanism: History, theory, implications. In Progress in Oceanography (Vol. 216). Elsevier Ltd. https://doi.org/10.1016/j.pocean.2023.103056

Pisano, A., Buongiorno Nardelli, B., Tronconi, C., and Santoleri, R.: The new Mediterranean optimally interpolated pathfinder AVHRR SST Dataset (1982–2012), Remote Sensing of Environment, 176, 107–116, https://doi.org/https://doi.org/10.1016/j.rse.2016.01.019, 2016.

[Figure]

Figure 1: Comparison of modelled (EV) and observed (OBS) thermal extreme events in the Adriatic-Ionian subdomain. Panels (a-b) represent respectively the observed and modelled time series alongside with the identification of the 10th and 90th daily percentiles for the period (here computed as a moving average within a 15-day sliding window); (c) highlights the events found in either series and their intensity; (d) compares the yearly cumulative intensity of the extreme events, and (e-f) compares for each month the observed and modelled number of MHWs and CSs respectively).

[Figure]

*Figure 2: Same as Figure 1, with reference to Dalmatian Islands subdomain*

[Figure]

*Figure 3: Same as Figure 1, with reference to Deep Adriatic subdomain*

[Figure]

Figure 4: Same as Figure 1, with reference to Kvarner Bay subdomain

[Figure]

*Figure 5: Same as Figure 1, with reference to Middle Adriatic subdomain*

[Figure]

*Figure 6: Same as Figure 1, with reference to Northern Adriatic subdomain*

[Figure]

*Figure 7: Same as Figure 1, with reference to Western Coast subdomain*

[Figure]

Figure 8: Same as Figure 1, with reference to the whole domain

---

## Author Response (AR1)

**RC1, 'Comment on egusphere-2024-1468', Anonymous Referee #1, 18 Jul 2024.**

The manuscript mostly concentrated on the Evaluation Run and its statistical properties by comparing the ocean model output with the available observations (SST, SLP, T). However, the manuscript missing a detailed analysis of the climate runs which is the main promise of the manuscript. The statistical analysis made for the Evaluation Run (Atmospheric forcings, Sea level variability and circulation patterns, Thermohaline properties, Extreme thermal events) should be repeated for all the climate runs.

*In this work the model validation was given a large attention as the idea was not only to assess the validity and limitations of the finding on the climate part, but also to provide some guidance on the use of the ensemble dataset associated with the present paper. Nonetheless, we concur on the opportunity of having a broader view on the climate results and we have expand this section with particular reference to sea surface temperature, salinity, and surface heat fluxes statistics (Section 3.2 -now also renamed into "Climate historical runs and projected climate change signal").*

*In the framework of this expansion it is important to notice (and we made it clearer in the revised version, see for instance lines 99-105 in the track change file) that the methodology for evaluation runs does not fully apply to the historical runs. While the former are driven by a reanalysis, which means that the RCM forcing the ocean model receives as boundary conditions reanalysis fields constrained to the observed atmospheric variability, the latter are driven by GCM fields, which result from a free run incorporating the observed variability "only" in terms of radiative fluxes (consistently with the constraints that can/cannot be controlled in a climate perspective) and generating the internal atmospheric variability as a free evolution under this condition. As a consequence, while atmospheric variability in reanalysis-driven runs is synchronised with the observed variability (and therefore it is licit to perform a direct one-to-one comparison of model and observations – that is, for instance, modelled fields for one day should match observed data for the same day), in GCM-driven runs this is only valid in statistical terms, and the modelled variability is expected to only match the statistical properties of atmospheric processes corresponding to a given radiative forcing regime. This is the reason why some assessment of the performance of the climate runs was actually included in the first version (Figures 13 and 16, now 14 and 21) only in aggregated terms.*

*On this ground, and with an eye on keeping the manuscript within a reasonable length, in the revised version we will expand along the following lines:*

- *Comparison of wind statistics in the climate run in CTR conditions against observations (Figure 5)*
- *Seasonal ensemble variations (percentiles) between SCE and CTR conditions in sea surface temperature (Figure 15), salinity (Figure 16) and net surface heat fluxes (Figure 17).*
- *Differences in the statistical distribution of sea surface temperature for the different subdomains (Figure 18), particularly in the perspective of supporting the interpretation of the results in terms of thermal extremes (see also comments by Rev. 2 and our response).*

As it is known, the open boundary conditions (OBCs) for the ocean models are critical, especially in the Adriatic Sea, the small differences in the salinity and temperature specified at the OBCs significantly affect the dense water formation and physical properties. In the manuscript, how

the OBC data were generated to force the ROMS model is not clear, need to be specified the methods and justified that could be used safely in a climate model.

*We thank the reviewer for pointing this out. In the revised version we have expanded the "model setup" Section with a more detailed explanation of how we imposed the boundary conditions and their modulation in the future scenario, and some comments on their suitability for the purposes of this study (see lines 118 to 143 in the track change version). Flanking the description of the methodology adopted we include a new figure (Figure 2) showing potential temperature and salinity distributions along the boundary cross section together with the velocity contours, also considering the EV\* run. In order to check that the prescribed climatological variations are realistic, we also included a comparison of the average trends, finding that the multidecadal tendency for potential temperature in the climate run in the historical period is well bracketed between the evaluation values (namely, the CMEMS reanalyses used as boundary conditions), while salinity trends (less clear in the reanalysis) appear slightly underestimated in the climate runs. The thermohaline properties appear consistent with typical values from the literature, particularly in terms of Modified Levantine Intermediate Water (MLIW, see for instance Bonaldo et al., 2016, and references therein), now more generally identified as Eastern Intermediate Water (EIW, Schroeder et al., 2024). The known cyclonic flow across the boundary cross section is weaker in EV than in EV\*, but is restored internally as geostrophic circulation (Figure 7), restoring the typical climatological values for MLIW around 0.10 m s$^{-1}$ (Orlić et al., 1992; Artegiani et al., 1997). Furthermore, the subdomain-based analysis of the evaluation run in the Results section has been expanded focusing on the surroundings of the model domain boundary (red polygon in Figure 1), and the Taylor diagram (Figure 11) now includes an assessment of the model in that region, showing a good agreement with measurements, with skill metrics comparable with the AdriSC reference (Pranić, et al., 2021).*

**RC2: 'Comment on egusphere-2024-1468', Anonymous Referee #2. Citation: https://doi.org/10.5194/egusphere-2024-1468-RC1**

Review of the paper

The manuscript deals with the present and the end-of-century, kilometre-scale ensemble modelling approach for the description of ocean processes in the Adriatic Sea using and ensemble of climate runs in a severe RCP8.5 scenario forced by the SMHI-RCA4 Regional Climate Model driven by CMIP5 General Climate Models as well as evaluation runs for the 1987-2010 period.

The text is well written and results are presented clearly.

*We thank the reviewer for this positive comment*

The authors show that the main behaviour of the model used is ''satisfactory''. However scenario simulations show results that necessitate a deeper investigation of the role of the model set up and of the forcing. The choice of the lateral boundary conditions is also of a crucial importance.

A deeper investigation of the relatively high theta and S changes in the deep Adriatic is recommended. This is also the case of the weak change of MHWs shown.

*We do appreciate the reviewer's suggestion. In fact, our positiveness lies mostly in the fact that the model performance against observations is comparable with the one exhibited by a state-of-the-art hindcast (AdriSC, referenced in the manuscript), which is a very good result in a climate model. The purpose of this manuscript and of the associated dataset is to pave the way to a number of studies on the many processes that take place in the Adriatic Sea. In principle the role of different factors (and in particular of atmospheric forcings and boundary conditions) depends on the process to be investigated (for instance, a study on river plume spreading in future conditions would not depend on the same processes and metrics as a study on marine heat waves or on dense water formation), and an extensive discussion on each of these aspects is beyond the scope of this manuscript and probably unfeasible for a single paper (see for instance the comment added in Lines 538-541 of the track change version). In this direction, the scope of the validation presented in the manuscript lies mostly in presenting the potential of the model and its dataset, the possible limitations, and in discussing the results presented in the climate scenarios. In the revised version we strove to better clarify the objectives of this paper and the necessary steps to be undertaken in the future applications.*

*Nonetheless, we agree with the reviewer that a somewhat deeper (though with an eye on keeping the manuscript to a reasonable size) general discussion on boundary conditions, atmospheric forcings, their role on the results and the implications for other applications would be beneficial for the paper and helpful in the use of the dataset, and we will follow the reviewer's suggestion. In the revised version we included in the "Model Setup" section (Lines 118 to 143 in the track change version) a more detailed explanation of how the boundary conditions were introduced and a check on the trend of the values prescribed in the climate simulations (reflecting the variability of the CMCC-CM profiles) against the ones prescribed in the EV and EV\* runs (reflecting the two versions of the CMEMS reanalysis), while a quantitative assessment of their quality is introduced in the Results in an updated version of the Taylor diagrams (see Figures 2 and 11 in the revised version). We point out that the thermohaline properties appear consistent with typical values from the literature, particularly in terms of Eastern Intermediate Water - Modified Levantine Intermediate Water (EIW or MLIW, see Schroeder et al., 2024, Bonaldo et al., 2016, and references therein), and the known cyclonic flow across the boundary cross section is weaker in EV than in EV\* but in any case is recreated internally as geostrophic circulation (see also response to Rev. 1), restoring the typical climatological values for MLIW around 0.10 m s$^{-1}$ (Orlić et al., 1992; Artegiani et al., 1997). Furthermore, the Taylor diagram (Figure 11) shows a good agreement with measured values also in the surroundings of the boundary, again with skill metrics comparable with the AdriSC reference (Pranić, et al., 2021).*

*In addition, also in the direction of expanding the "climate" part of the manuscript as suggested by Rev. 1, we introduced some additional results from the climate runs.*

*An assessment of the wind regimes in the historical part of the climate runs (Figure 5) shows a good match with observations at sea in the Northern Adriatic also for the GCM-driven simulations, suggesting that overall realistic wind regimes are used also as a forcing for the climate runs.*

*The variations of ensemble seasonal percentiles between SCE and CTR conditions for sea surface temperature, salinity, and net surface heat fluxes (Figures 15 to 17), as well as the differences in the daily climatological CDFs for SST (Figure 18 – the equivalent for heat fluxes is less informative and has been omitted) in the different subdomains are introduced mainly to*

*enhance the analysis of the climate variability and the discussion of the results on thermal extremes. Nonetheless, complemented with the trends on the boundary conditions (Figure 2, panels c and f), they have been used to draw some considerations on the effect of local dynamics, river runoff and boundary conditions in the basin properties (see for instance Lines 478-484 in the track change version), although again recalling that fully disentangling the role of each factor in general is beyond the scope of this work.*

*Elements along these lines have been introduced throughout the discussion in order to address the reviewers questions raised above, with all the single modifications visible in track change, particularly focusing on temperature changes and thermal extremes (less information could actually be found for salinity, at least for this purpose), the underlying assumptions, and the open challenges.*

I recommend major revisions.

Specific comments:

-15. '' with particularly encouraging results''

Could authors explain to what extent results are encouraging?

*The overall good capability of the model to reproduce the main features of observed Marine Heat Waves (MHWs) and Cold Spells (CSs), such as timing, intensity, and interannual variability suggests that our dataset could effectively be used for studies involving thermal extremes (e.g. linked to ecological processes, etc.). We rephrase this to make it clearer in the revised version (Lines 23-25 in the track change version).*

-45. '' (?Denamiel et al., 2021a) ''

Please correct if needed.

*Thanks, fixed*

-50. '' ... ranging from the very evolution of the global...''

Could authors verify this sentence?

*Thanks, modified into "...from the evolution of the global climate to how this signal propagates through different scales, and how the adopted numerical description impacts the final results". Lines 64-65 in the track change version.*

-100. ''Potential temperature (θ), salinity (S), momentum...''

Could authors describe how momentum is used in the boundary condition set up?

*In the revised version we will provide more details on the boundary conditions. In particular, we will point out that we imposed Chapman conditions (Chapman, 1985) for free surface, Flather conditions (Flather, 1976) for 2D momentum components, and nudged radiative conditions for 3D momentum components and tracers (potential temperature and salinity).*

-105. ''...were modulated accordingly with the anomalies computed from Med-CORDEXderived CMCC-CMprofiles (Scoccimarro et al., 2011) in the norhtheasternmost grid cell of the Ionian Sea.''

Could authors better describe the approach followed?

*Thanks, in the revised version we reshaped the description along these lines: "For the EV run, daily reanalysis values were directly interpolated on the model grid points throughout the cross section. For the climate runs, climatological monthly values were first computed from the reanalysis fields with reference to the 1987-2017 period. These values were then perturbed with the anomalies computed, with reference to the same period, from Med-CORDEX derived CMCC-CM profiles (Scoccimarro et al., 2011) in the northeasternmost grid cell of the Ionian Sea." See Lines 128-131 in the track change version.*

-185. ''Furthermore, although being the only available option the evaluation of SMHI-RCA4, ERA-INTERIM known to be far from the "perfect boundary conditions" hypothesis, particularly in terms of rainfall-related quantities (Bao and Zhang, 2013).''

This sentence is rather unclear, could authors rephrase it.

*We thank the reviewer for pointing this out, actually we were missing a verb! We apologise for that. We add the verb and slightly modify the sentence into "Furthermore, although being the only available option the evaluation of SMHI-RCA4, ERA-INTERIM does not presently represent the state of the art for atmospherical modelling, and is known to be far from the "perfect boundary conditions" hypothesis, particularly in terms of rainfall-related quantities (Bao and Zhang, 2013)". Line 222, track change version.*

-210. Whereas the comparisons shown in Fig.2, Fig.3 and 4 show that the model behaviour is rather satisfactory as stated by the authors: ''thus performing significantly better than most of the RCMs available for this geographical area'', it would be interesting to illustrate this by one or two concrete examples.

*We thank the Reviewer for helping us noticing that our reference to the RCA4 regional climate model (RCM) skill were probably slightly overenthusiastic. We have refined this phrase removing "thus performing significantly better than most of the RCMs available for this geographical area" from the original sentence. In fact, strictly speaking this actually represents an overstatement since a specific assessment based on a multi-RCM comparison over this particular region and variables lies outside the scope of the study. Nevertheless RCA4 shows overall representative skills for essential climate variables as preliminarily assessed in the context of a previously published article and involving similar geographical domain (Bonaldo et al., 2023) as well as in review articles, including the large CORDEX ensemble (Coppola et al., 2021; Diez-Sierra et al., 2022; Vautard et al., 2021), and specifically over the Adriatic region where Belušić Vozila et al.,*

*(2019) consider wind climate variable specifically. These considerations and the related references have been added to the revised version of the manuscript (lines 252-256 in the track change version).*

*Another element driving us towards using this model is that at the time in which climate simulations were gathered, RCA4 was the RCM with the largest number of simulations (corresponding to different driving GCMs) with a sub-daily time frequency for quite a large number of variables, required for driving the ROMS model. As outlined in the methodology section, this approach is appealing within the research framework, as it aims to limit uncertainty sources by focusing on a single RCM setup rather than exploring multiple RCMs driven by different GCMs.*

-230. "This northbound improvement of the model skills suggests that internal dynamics partially compensate for the missing variability component in the boundary conditions." ; "…the fairly good performance on the Northern Adriatic coast permits a more straightforward use in this region, also in terms of boundary conditions for local applications".

-What is the sampling interval used in Fig.5?

-Are tidal oscillations included in Fig .5?

-The tidal amplitude is known high in the northern Adriatic; could authors discuss the impact of the tidal amplitude of the model performance shown in Fig.5.

-Again authors should further discuss the lateral boundary conditions.

*Although tides were included in the EV and EV\* runs by considering 15 tidal components from the TPXO dataset (Egbert and Erofeeva, 2002), Figure 6 (previously Figure 5) refers to daily-averaged data, and therefore semi-diurnal and diurnal components are lost, and the underestimate in sea surface level variability shown in that Figure does not include tidal variability. In a deeper analysis not included in this manuscript for the sake of synthesis we found that tidal variability resulting from TPXO is generally underestimated. This may contribute to some mismatch in circulation and tracer transport patterns over the short term, but since the result of the validation is considered in aggregated terms, we don't see obvious reasons to expect systematic errors introduced by this factor, whilst it most likely contributes to add some noise around the average skills.*

*This is better explained in the revised version, at lines 270-277 in the track change version.*

-270. "… , suggesting that SST does not show any macroscopic sign of a spurious drift related to the model implementation ".

 Could authors rephrase this sentence?

*Thanks, in this form this sentence aimed at addressing the (undeclared) possibility that a bias in the flux parameterizations at the air-sea interface or numerical issues could, at the multi-decadal time scale, result in an unrealistic temperature drift. In fact, for this sentence to be clear it would call for a better discussion of this possibility, but then it would probably burden the discussion without adding an important contribution (in the end, the drift is not there!). We thus decided to remove this sentence.*

-280. '' In turn, while intermediate to high S values are mostly well reproduced, low to mid salinity tends to be overestimated, particularly in the Kvarner Bay and in the Dalmatian Islands.''

Please better discuss and explain the overestimation of the lowest S values.

*Thanks, two main aspects can have a role in this result. First, the estimate of submarine freshwater inputs in the karstic northeastern and eastern Adriatic coast is a recognisedly challenging task in the area, and can lead to significant uncertainties. Secondly, the model resolution does not permit a complete description of the complex geomorphology of that coast, and therefore of its small-scale circulation patterns. This is better explained in the revised version, lines 335-341 in the track change document.*

-285. '' This suggests that the climate ensemble, whose implementation began before the release of the latest version of MFS (Escudier et al., 2020), should not be considered prone to major elements of obsolescence associated with the use of a previous dataset (Simoncelli et al., 2019)''.

Could authors explain how this can be deduced from Fig.10.

*Here the idea is that, if the use of the latest version of MFS does not significantly improve the overall skill metrics, it seems reasonable to expect that the use of a previous dataset (the only one available at the time of the ensemble implementation) to compute the climatologies at the boundary should not lead to major shortcomings in the climate runs. We better frame this concept in the revised version (lines 351-354 in the track change document).*

-305. '' In the deep Adriatic, an apparent tendency to underestimate average values of θ and S throughout the year is actually the result of some shortcomings in the description of thermohaline properties in the upper layers.''

The sentence needs further explanation of the mentioned shortcomings.

*Thanks for this suggestion, in fact the sentence should be adjusted. More precisely, the first part refers to panel c, but we do not have enough elements to actually attribute this result to some processes, or shortcomings taking place in the upper layers. Instead, while there is certainly an overestimate of heating and mixing in the upper layers, the observed underestimate at deeper layers could be inherited from the dataset used for initialization and boundary conditions (the results presented by Pranic et al. 2021, which were based on the same datasets, showed very similar values). We discuss, and hopefully better clarify, this in the revised version at lines 369-384 (track change document), though pointing out that a conclusive interpretation of this mismatch requires a dedicated effort.*

-370. Please correct : ''hle200 m''

*Thanks, adjusted.*

-375. ''Below the upper layer, θ increase varies from + 2.8°C for h=200 m to +1.3°C for h≥800 m, S increase varies from +0.21 to +0.17, and σθ varies between-0.44 and-0.15 kg m−3.''

Authors should discuss the relatively high values of salinity and theta changes in the deep Adriatic (shown in Fig.15c) and present comparison with results from previous work. Also, why two among the vertical profiles of the theta change are truncated at depths less than ~630 m, (Fig. 15c left).

*The truncation of the bottom values of the ensemble spread resulted from the graphical setting for the x-axis limits, this has been adjusted in the revised version (Figure 20). The values of the temperature and salinity variations in the Deep Adriatic are consistent with the trends prescribed at the boundary (Figure 2) and a comment along this line has been added in the revised version (track change, lines 478-484).*

-385. ''Under this approach, modelled differences between SCE and CTR conditions (expressed as monthly mean cumulative intensity of the events) appear generally minor and in any case only occasionally statistically significant.''

Here also, authors should mention results from previous work, if available.

*Thanks, this has been discussed in the light of trends from the recent past in the literature (lines 494-496 of the track change document) and flanked with some further comments on the implications of the choice of the reference thresholds for MHW and CS definition (lines 489-491 in the track change version) and on the effect of changes in the SST statistical distribution (Figure 18, lines 500-502, and 510-512)*

**References**

Artegiani, A., Bregant, D., Paschini, E., Pinardi, N., Raicich, F., Russo, A., 1997a. The Adriatic Sea general circulation. Part I: air–sea interactions and water mass structure. Journal of Physical Oceanography. 27, 1492–1514.

Belušić Vozila, A., Güttler, I., Ahrens, B., Obermann-Hellhund, A., & Telišman Prtenjak, M. (2019). Wind Over the Adriatic Region in CORDEX Climate Change Scenarios. *Journal of Geophysical Research: Atmospheres*, *124*(1), 110–130. https://doi.org/10.1029/2018JD028552

Bonaldo, D., Bellafiore, D., Ferrarin, C., Ferretti, R., Ricchi, A., Sangelantoni, L., & Vitelletti, M. L. (2023). The summer 2022 drought: a taste of future climate for the Po valley (Italy)? *Regional Environmental Change*, *23*(1). https://doi.org/10.1007/s10113-022-02004-z

Coppola, E., Nogherotto, R., Ciarlo', J. M., Giorgi, F., van Meijgaard, E., Kadygrov, N., et al. (2021). Assessment of the European Climate Projections as Simulated by the Large EURO-CORDEX Regional and Global Climate Model Ensemble. *Journal of Geophysical Research: Atmospheres*, *126*(4), 1–20. https://doi.org/10.1029/2019JD032356

Diez-Sierr, J., Iturbide, M., Gutiérrez, J. M., Fernández, J., Milovac, J., Cofiño, A. S., et al. (2022). The Worldwide C3S CORDEX Grand Ensemble A Major Contribution to Assess Regional Climate Change in the IPCC AR6 Atlas. *Bulletin of the American Meteorological Society*, *103*(12), E2804–E2826. https://doi.org/10.1175/BAMS-D-22-0111.1

Chapman, D.C., 1985. Numerical treatment of cross-shelf open boundaries in a barotropic coastal ocean model. Journal of Physical Oceanography 15, 1060‑1075.

Bonaldo, D., Benetazzo, A., Bergamasco, A., Campiani, E., Foglini, F., Sclavo, M., Trincardi, F., Carniel, S. (2016). Interactions among Adriatic continental margin morphology, deep circulation and bedform patterns. *Marine Geology*, *375*. https://doi.org/10.1016/j.margeo.2015.09.012

Flather, R.A., 1976. A tidal model of the northwest European continental shelf. Memoires de la Society Royal des Sciences de Liege 6, 141‑164.

Orlić, M., Gacić, M., La Violette, P.E., 1992. The currents and circulation of the Adriatic Sea. Oceanologica Acta 15 (2), 109–124.

Pranić, P., Denamiel, C., & Vilibić, I. (2021). Performance of the Adriatic Sea and Coast (AdriSC) climate component - A COAWST V3.3-based one-way coupled atmosphere-ocean modelling suite: Ocean results. *Geoscientific Model Development*, *14*(10), 5927–5955. https://doi.org/10.5194/gmd-14-5927-2021

Schroeder, K., ben Ismail, S., Bensi, M., Bosse, A., Chiggiato, J., Civitarese, G., Falcieri, F. M., Fusco, G., GaČiĆ, M., Gertman, I., Kubin, E., Malanotte-Rizzoli, P., Martellucci, R., Menna, M., Ozer, T., Taupier-Letage, I., Vargas-Yanez, M., Velaoras, D., & Vilibic, I. (2024). A consensus-based, revised and comprehensive catalogue for Mediterranean water masses acronyms. *Mediterranean Marine Science*, *25*(3), 783–791. https://doi.org/10.12681/mms.38736

Schumacher, D. L., Singh, J., Hauser, M., Fischer, E. M., Wild, M., & Seneviratne, S. I. (2024). Exacerbated summer European warming not captured by climate models neglecting long-term aerosol changes. Communications Earth and Environment, 5(1). https://doi.org/10.1038/s43247-024-01332-8

Scoccimarro, E., Gualdi, S., Bellucci, A., Sanna, A., Fogli, P. G., Manzini, E., Vichi, M., Oddo, P., and Navarra, A.: Effects of Tropical Cyclones on Ocean Heat Transport in a High-Resolution Coupled General Circulation Model, Journal of Climate, 24, 4368 – 4384, https://doi.org/10.1175/2011JCLI4104.1, 2011.

Vautard, R., Kadygrov, N., Iles, C., Boberg, F., Buonomo, E., Bülow, K., et al. (2021). Evaluation of the Large EURO-CORDEX Regional Climate Model Ensemble. *Journal of Geophysical Research: Atmospheres*, *126*(17). https://doi.org/10.1029/2019JD032344